# Environmentally triggered shifts in steelhead migration behavior and consequences for survival in the mid-Columbia River

**Jared E. Siegel**[1]*, **Lisa G. Crozier**[2], **Lauren E. Wiesebron**[1¤], **Daniel L. Widener**[1]

**1** Ocean Associates, Under Contract to the Northwest Fisheries Science Center, National Marine Fisheries Service, NOAA, Seattle, WA, United States of America, **2** Northwest Fisheries Science Center, National Marine Fisheries Service, NOAA, Seattle, WA, United States of America

¤ Current address: Estuarine and Delta Systems, Royal Netherlands Institute for Sea Research, Korringaweg, Yerseke, The Netherlands
* Jared.Siegel@noaa.gov

**Data Availability Statement:** All relevant data are within the manuscript and its Supporting Information files.

## Abstract

The majority of Columbia River summer-run steelhead encounter high river temperatures (near or > 20˚C) during their spawning migration. While some steelhead pass through the mid-Columbia River in a matter of days, others use tributary habitats as temperature refuges for periods that can last months. Using PIT tag detection data from adult return years 2004–2016, we fit 3-component mixture models to differentiate between "*fast*", "*slow*", and "*over-wintering*" migration behaviors in five aggregated population groups. *Fast* fish migrated straight through the reach on average in ~7–9 days while *slow* fish delayed their migration for weeks to months, and *overwintering* fish generally took ~150–250 days. We then fit covariate models to examine what factors contributed to the probability of migration delay during summer months (*slow* or *overwintering* behaviors), and to explore how migration delay related to mortality. Finally, to account for the impact of extended residence times in the reach for fish that delayed, we compared patterns in estimated average daily rates of mortality between migration behaviors and across population groups. Results suggest that migration delay was primarily triggered by high river temperatures but temperature thresh-olds for delay were lowest just before the seasonal peak in river temperatures. While all pop-ulations groups demonstrated these general patterns, we documented substantial variability in temperature thresholds and length of average delays across population groups. Although migration delay was related to higher reach mortality, it was also related to lower average daily mortality rates due to the proportional increase in reach passage duration being larger than the associated increase in mortality. Lower daily mortality rates suggest that migration delay could help mitigate the impacts of harsh migration conditions, presumably through the use of thermal refuges, despite prolonged exposure to local fisheries. Future studies track-ing individual populations from their migration through reproduction could help illuminate the full extent of the tradeoffs between different migration behaviors.

**Funding:** The author(s) received no specific funding for this work.

**Competing interests:** The authors have declared that no competing interests exist.

## 1. Introduction

Pacific salmon and anadromous trout populations (*Oncorhynchus* spp.) have greatly declined from historical levels [1] and remaining populations face major threats from habitat degradation/loss [2] and climate change [3]. Phenotypic and genetic diversity within and among populations can increase population resilience to environmental change and the stability of dependent fisheries [4]. Unfortunately, the overall genetic and phenotypic diversity of salmon and anadromous trout has been reduced by the extirpation of populations [5], habitat loss, fishing, and hatchery practices [6]. These impacts have left many remaining populations more vulnerable to climate change as their ability to adapt has been diminished [3]. Accordingly, it is a management priority to protect the remaining life history diversity and the range of habitats that support this diversity [7].

Steelhead trout (*Oncorhynchus mykiss*), the anadromous form of rainbow trout, demonstrate a wide array of life history strategies. For example, they exhibit large variation within- and across-populations in the number of years spent in freshwater as juveniles and in saltwater before returning to spawn [8]. In a few populations, generally immature "half-pounders" return to freshwater after a few months in the ocean to feed and overwinter before migrating back to saltwater the following spring [9]. Unlike other Pacific salmon species, steelhead have the potential to return to the ocean after spawning and re-ascend rivers to spawn again in subsequent years ("repeat spawners") [10, 11], and in some populations fish may spawn numerous times [12]. Populations can include co-occurring anadromous and resident life history strategies [13] with the expression of anadromy responding to facultative [14] and evolutionary pressures [15].

Steelhead also demonstrate substantial variation in adult migration behaviors. They typically spawn in the spring, though steelhead in many rivers return to freshwater during all months of the year. Although steelhead demonstrate a continuous spectrum of adult migration behaviors, they are generally categorized into two ecotypes in the contiguous United States. Those that begin maturing in the ocean and arrive to freshwater between November and April are referred to as "winter-run", while those that return between May and October and sexually mature in freshwater are referred to as "summer-run" [16]. Summer-run steelhead may spend up to 9 months in freshwater before spawning. While winter-run populations are more numerous, summer-run populations are predominant in mountainous-inland basins which require long migrations to access [16] and often occur in tributaries that historically had migration barriers at high winter flows [17].

Columbia River steelhead populations within the Interior Columbia Recovery Domain (which includes populations located east of the Cascade Mountain crest) are almost exclusively summer-run and are all listed as threatened under the U.S. Endangered Species Act [18]. Today these populations face many challenges during their upstream migrations that can influence their reproductive success. Once fish enter the river they must find and navigate a series of fish ladders to pass numerous hydroelectric dams. During their migration steelhead are targeted by growing numbers of pinnipeds (seals and sea lions) in the lower river [19–21] and sport and tribal fisheries throughout the Columbia River. Additionally, summer-run steelhead arrive in the Columbia River primarily during mid-summer, from July through September, when mainstem river temperatures generally peak above 20°C and commonly exceed 21°C, a level that can deter migration [22].

While some summer-run steelhead migrate directly to spawning tributaries, a large proportion delay their migration by holding in mainstem or tributary habitats. Diversity in migration behaviors acts as bet hedging to ensure that some portion of the population survives to spawn given environmental variability that may favor different behaviors at different times [23]. The

likelihood of migration delay appears to be largely environmentally triggered as fish seek temperature refuges in tributary habitats and cold-water plumes during periods of high mainstem river temperatures [24–26]. Migration delays can last weeks to months before fish continue moving upstream to natal tributaries ahead of spawning in the spring. This behavior occurs prominently in the Columbia River reach from Bonneville Dam to McNary Dam (hereafter mid-Columbia) [24, 25], which contains numerous mountain-fed, cold-water tributaries.

The challenge of reproduction for summer-run steelhead includes both spatial (migration from the ocean to spawning grounds) and temporal (survival in freshwater from the arrival in the summer to spawning in the spring) components. Since summer-run steelhead reduce consumption of prey upon returning to freshwater they have a limited supply of energy to migrate, sexually mature, and spawn. As such, it may benefit some fish to delay their migration in the mid-Columbia during the summer if mainstem conditions are strenuous or if refuge habitats in the mid-Columbia are superior (generally cool and deep) compared to those accessible further upstream or in spawning tributaries. However, despite the potential benefits, migration delays have previously been associated with lower survival through this river reach, which may be a consequence of increased exposure to the large local tribal and sport fisheries [27]. Previous analysis of tag detections at dams suggests that the majority of migration mortality of summer-run steelhead within the Columbia River hydropower system occurs in the mid-Columbia, with mortality rates generally around 20% [28].

In this investigation we 1) examined patterns in upstream migration delays in the mid-Columbia between Bonneville and McNary dams, 2) investigated drivers of this behavior, and 3) related this behavior to overall reach-specific survival and daily mortality rates. We focused on the mid-Columbia reach because this is where fish most commonly seek temperature refuge [25], where fishing pressure is high, and because it is a shared migration reach for many inland populations. To achieve these objectives, we used data from steelhead implanted with passive integrative transponder (PIT) tags as juveniles from adult return years 2004–2016. This multi-year time-series captured a wide range of inter-annual environmental variability, allowing us to explore how summer-run steelhead behavior and survival responded to environmental gradients and how responses varied across populations. Results from this investigation can be used by fisheries managers to help mitigate mortality to steelhead populations of concern and to plan for impacts from climate change.

As past research has suggested, we hypothesized that all steelhead populations would demonstrate increased probabilities of delay during stressful migration conditions, primarily as a response to high temperatures. Nevertheless, we expected population-specific variation in the propensity to delay as those with higher quality holding habitat in natal tributaries, or access to other refuge habitats further upstream, might be less likely to delay in the study reach. In addition, we expected that populations that were less likely to delay would demonstrate lower overall reach mortality. However, because overall reach mortality does not account for time-dependent mortality, it might overestimate the benefits of rapid migration. Therefore, we included a time-dependent mortality rate in our analysis of reach-survival. We hypothesized that mortality rates of rapid migrants and delayed migrants would differ less or possibly demonstrate the inverse of the relationship for overall reach survival.

## 2. Materials and methods

### 2.1. Dataset

We queried the Columbia Basin PIT-Tag Information System database [29] for records of summer-run steelhead that returned as adults from 2004 to 2016, which included known-origin hatchery and wild fish tagged as juveniles between 2000 and 2015. We only considered fish

released upstream of McNary Dam as juveniles, and we inferred population origin from release sites. We determined that a fish was a migrating adult if it was detected in an adult fish ladder at least one year after the smolt migration year. We excluded repeat spawners, which were very rare in our dataset (<1%), by using only the first adult migration in our analyses. In total, we used information from 43,495 returning adults.

The large majority of hatchery fish in the dataset were tagged at hatcheries prior to juvenile release. Wild fish were primarily tagged at rotary screw traps, which aim to randomly sample individuals from watersheds as they move downstream, and during active sampling (e.g., seine netting and electrofishing). Further details of database development and processing are provided in a prior report [28]. All subsequent data processing and analyses were performed using the statistical program R [30].

## 2.2. Population groups

We grouped summer-run populations for analysis based on similarities in adult migration timing and migratory pathways (Fig 1 and S1 Table). The Upper Columbia and Middle Columbia population groups align largely with Distinct Population Segments (DPS) defined

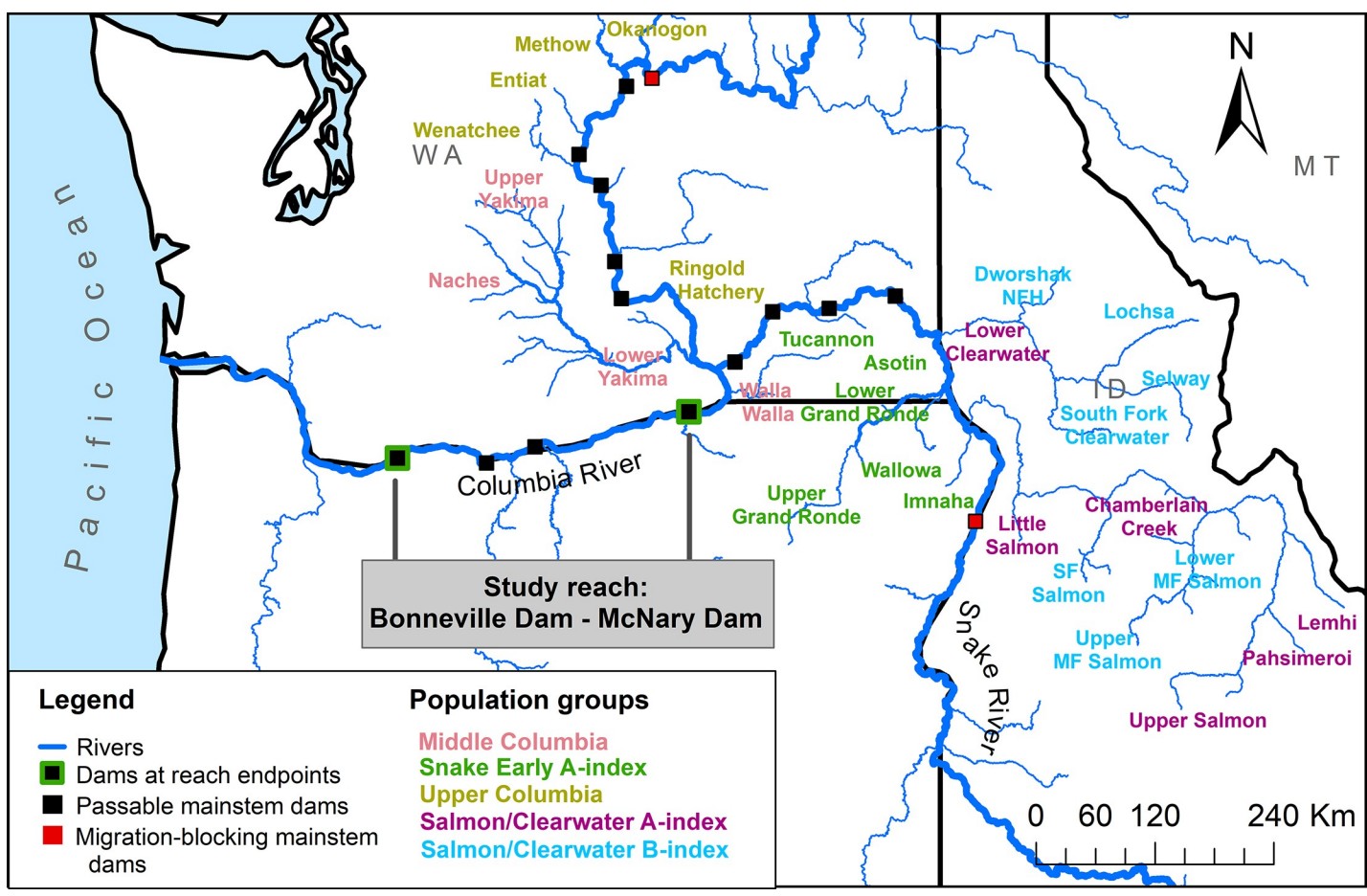

**Fig 1. Study area map.** Map of study area showing the study reach and the location of each population within each population group. Salmon/Clearwater A- and B-index populations are labeled by whichever group represented the majority of tags within the tributary (some hatchery programs release fish with genetics from one life-history strategy into populations that are primarily composed of the other). While Pahsimeroi River steelhead are generally considered A-index, our dataset contained a significant number of tags from hatchery-raised B-index stock that are released in the basin.

for listing under the U.S. Endangered Species Act [18]. Note that our Middle Columbia group did not include populations in the DPS with confluences downstream from McNary Dam.

We split the summer-run steelhead from the Snake River DPS into three groups to better represent the extensive life-history variation within this large DPS. This resulted in the Snake Early A-index group, composed of early arriving populations, and the Salmon/Clearwater (Sal/Clear) A-index and B-index groups respectively. For harvest management purposes Snake River summer-run steelhead have been separated into two migration groups based on characteristics that can be determined at harvest; A-index fish are smaller (< 78mm), and arrive at Bonneville Dam earlier compared to B-index fish. Populations that are considered primarily B-index are limited to a few populations in the Salmon and Clearwater basins [18, 31]. However, size and timing criteria do not correspond perfectly to population of origin because there is substantial overlap in these characteristics [11, 31]. Thus, our A- and B-index designations represent the most-likely designation within a population. The identification of life history type for separating out Sal/Clear A-index and B-index migration groups generally followed previously defined population designations [31], though we also incorporated information from hatchery releases in tributaries that targeted a phenotype. Sample sizes of population groups ranged from 1,806–22,792 fish (S1 Table) and the mean annual percent of hatchery origin ranged from 39–87% (S2 Table).

## 2.3. Migratory behavior designation

Summer-run steelhead demonstrated tri-modal distributions of log-transformed travel times, defined as days from first detection at Bonneville Dam to first detection at McNary Dam (Fig 2). These three modes were represented by "*fast*" fish which generally migrated directly through the reach within a week or two, "*slow*" fish which took anywhere from a couple of weeks to months to pass the reach, and the rarer "*overwintering*" fish which remained in the reach for hundreds of days or more. We note that while migration delay was most commonly exhibited in the study reach, some *fast* fish delayed in other reaches and *overwintering* behavior was actually more common upstream, though still comparatively rare.

For each population group, we fit a 3-component mixture model [32] to log-transformed travel times using the package mixtools [33] to assign a probability of migration behavior to

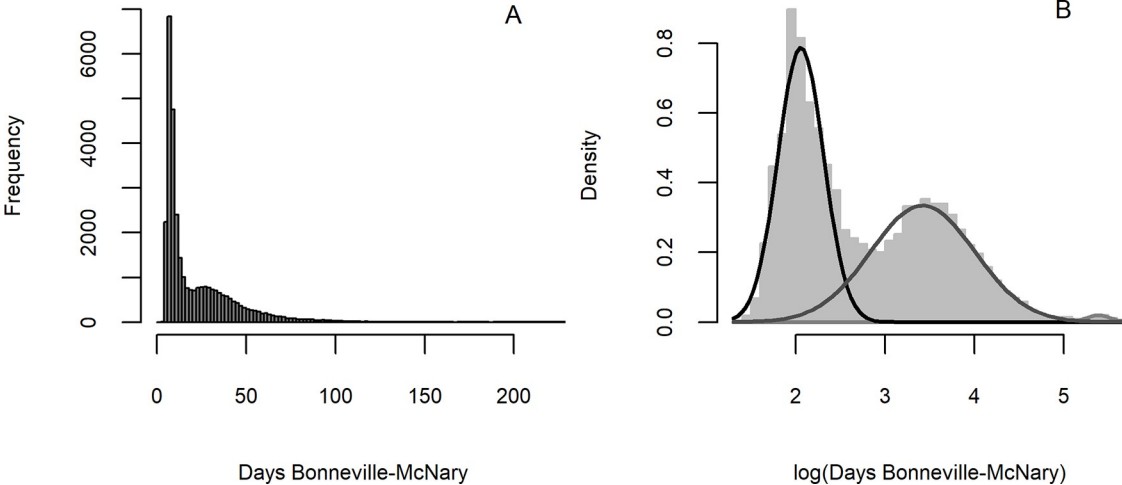

**Fig 2. Travel time distribution.** Histogram of mid-Columbia travel times through the study reach from Bonneville to McNary dam (A) and log-transformed travel times fit to a tri-modal Gaussian mixture model for *fast*, *slow*, and *overwintering* fish (B) for entire dataset.

each fish that was detected at both dams ($n$ = 32,740). The lower number of fish detected at both dams compared to our entire dataset ($n$ = 43,495) primarily reflects apparent survival as detection efficiencies at McNary Dam were near 100% during the study period [28]. The probability density function of the log travel times ($tt$) between Bonneville and McNary dams for a given population is a weighted composition of three distinct normal distributions for *fast*, *slow*, and *overwintering* behaviors:

$$\log (tt) \sim \lambda_{fast} \cdot N(\mu_{fast}, \sigma^2_{fast}) + \lambda_{slow} \cdot N(\mu_{slow}, \sigma^2_{slow}) + \lambda_{ow} \cdot N(\mu_{ow}, \sigma^2_{ow}), \tag{1}$$

where the $\lambda$'s are the component mixing weights, which sum to 1, and $\mu$'s and $\sigma^2$'s are the means and variances of the component normal distributions. An example of this model fit to the combined set of log travel times across populations provided a reasonable fit to the data (Fig 2B) and an improvement by AIC criteria compared to a 2 component model. The output of this model produces a probability of inclusion in each component distribution for every fish.

## 2.4. Travel times by migration behavior

To describe patterns in travel times for *fast*, *slow*, and *overwintering* fish respectively, we used the results from the mixture models to assign the corresponding behavior with the highest probability to each fish. We then used generalized additive mixed models (GAMMs) to describe seasonal patterns in average migration travel times for each behavior. We took this step to demonstrate that the average length of delays varied predictably across the summer, and to translate survival estimates into daily mortality rate estimates (section 2.7). We fit GAMMs to each population group/behavior combination separately using the package mgcv [34, 35]. We limited the number of knots for the smoothing splines to 6 to allow some flexibility but to prevent overfitting. The resulting model for the log travel time ($tt$) in days for fish $i$ was:

$$\log (tt_i) = \beta_0 + s(D_i) + \gamma_j + \epsilon_i, \tag{2}$$

where $\beta_0$ is the intercept, $s(D)$ is the smoothing spline for date of arrival at Bonneville Dam, $\gamma_j$ is the random effect for the year $j$ in which fish $i$ migrated, and $\epsilon_i$ is the residual error, where the residual errors were assumed to be normally distributed with mean zero and constant variance.

## 2.5. Probability of delay

We next used GAMMs to assess how environmental conditions and fish characteristics influenced the probability of a fish delaying their migration (expressing either *slow* or *overwintering* behaviors). We combined the probabilities for *slow* behaviors and *overwintering* because our focus was on the response to summer conditions, and it is not necessarily the case that those groups are determined at that time. The decision to *overwinter* may occur separately at a later date once a fish has already delayed. The *overwintering* group had a limited impact on our results because very few fish (~1%) fell into that category.

We considered the following environmental variables in models for the probability of delay: river temperature ($T$), river flow ($F$), dam spill volume ($S$), and the date of first detection at Bonneville Dam ($D$). The environmental variables ($T$, $F$, and $S$), which were accessed from the Columbia River Data Access in Real Time database [36], were averaged over the median *fast* fish travel time period (8 days) starting on the day of first detection at Bonneville Dam. These 8 days represent the period of exposure during which a fish makes the "decision" to

delay or to continue upstream. We used temperature and flow measurements from Bonneville Dam, which were very similar to measurements from other dams in the reach. However, we averaged spill across all dams (Bonneville, The Dalles, John Day, and McNary) due to greater variability in this variable, though measurements were still highly correlated on the daily level (Pearson's $r = 0.82$–$0.92$).

Only models with one or none of $F$, $S$, and $D$ were considered because these variables were highly correlated ($r > 0.7$). Spill tends to decrease over the summer with concurrent declines in flow. However, spill is generally abruptly shut off at the end of August. While temperature ($T$) is also related to $F$, $S$, and $D$ (though in a nonlinear fashion), we expected independent impacts from these variables and thus chose to consider them together in models.

While temperature directly impacts metabolism and physiological processes, flow and spill may impact migration by altering the physical requirements of dam passage and upstream migration. In contrast, variation in the probability of delay by arrival date might reflect population-specific evolved behavioral thresholds to account for variation in the quality of accessible holding habitat in each distinct migration route. To better interpret the combined effects of these related variables, we visually examined the regression surfaces formed between $T$ and $F$, $S$ or $D$ with the data.

We also considered two-level categorical (binary) variables for fish origin ($O$: hatchery vs wild), ocean age ($A$: 1 year vs 2–3 years), and juvenile transportation history ($J$: transported vs not), which applied to Snake River fish only. A portion of Snake River juveniles are barged from the Snake River dams to Bonneville Dam to avoid mortality in the hydrosystem during their migration to the ocean (see [37]). However, there is some evidence that juvenile barging can impact the upstream migration success of adults [38–40]. Finally, we considered a random effect for year.

We first fit a set of global models for each population group for the probability of delay (*pDelay)* for individual $i$, where *pDelay* is the combined probability of being *slow* or *overwintering* for each individual that was output from the mixture models represented by Eq 1. That is, we treated the predicted probabilities from the mixture models as the response. We assumed that *pDelay* is linear on the logit (log odds) scale. The following represents the general form of the largest possible model:

$$\text{logit}(pDelay_i) = \beta_0 + s(T_i) + s(X_i) + \beta_1 O_i + \beta_2 A_i + \beta_3 J_i + \gamma_j + \epsilon_i, \tag{3}$$

where $\beta_0$ is the intercept, $\beta_1$, $\beta_2$, and $\beta_3$ are parameters for the non-reference values of the categorical variables, $s(\cdot)$ are smoothing splines for the continuous variables, $X$ represents one of the correlated variables $F$, $S$, or $D$, $\gamma_j$ is a random effect for the return year $j$ in which individual $i$ migrated, and $\epsilon_i$ is the residual error, where the residual errors were assumed to be normally distributed with mean zero and constant variance. We limited splines to only 5 knots in these models due to the higher number of parameters compared to Eq 2.

We note that this equation represents the probability of delay for fish that survived. We make the simplifying assumption that the factors that impact delay are unbiased with regard to survival. In addition, while we included information from fish tagged by a wide variety of hatchery and wild fish tagging programs, some level of disproportionate representation is likely to remain given that programs had different protocols that often changed over time. Accordingly, we also made the assumption here and in subsequent survival models that tagged returning adults were representative within the unique combinations of categorical variables assessed within population groups (age/origin/juvenile transport groups).

After fitting global models for *pDelay* with all considered variables, we used the package MuMIn [41] to perform model selection retaining each model with the lowest AICc value. We

assessed the goodness of fit of selected models using the area under the receiving operator curve (AUC) [42]. In this case, the AUC represents the probability that a randomly selected fish that delayed migration (as defined in Eq 1) had a higher predicted probability of delay than a randomly selected fish that was designated as a *fast* migrant.

## 2.6. Behavior and survival

Fish that were detected at McNary Dam or at any other dam further upstream during migration were considered survivors (Fig 1), and the remainder were considered mortalities. This allowed us to use logistic regression to model the probability of survival through the Mid-Columbia reach (*pSurv*). Based on mark-release-recapture modeling by Crozier et al. [28], estimated detection probabilities were near 100% at McNary Dam in most years for upriver steelhead, averaging 97.2% and 97.7% for Upper Columbia and Snake River summer steelhead respectively. Including fish that were missed at McNary Dam but detected at other dams further upstream is likely to account for the majority of missed detections due to similarly high detection rates at upstream dams combined with consistently high migration survival through these reaches. Accordingly, this methodology produces nearly identical estimates of survival as estimated from mark-recapture modeling methods [28, compare to table 12].

For models of the probability of survival through the mid-Columbia reach (*pSurv*) we considered the predicted probability of delay ($\widehat{pDelay}$) from Eq 3 for the entire dataset (including identified mortalities) as our primary explanatory variable. We also included the categorical variables and the random effect for year from Eq 3 (*O*, *A*, *J* and *γ*) as these variables may have an impact on survival beyond their influence on delay. Finally, we considered a linear effect to account for annual fisheries exploitation. We calculated the harvest rate (*H*) as an annual proportion by dividing the total estimated fishing mortality of summer-run steelhead by the estimated total run size during the year from June to the end of October (data provided by Jeromy Jording, NMFS). Data for the non-tribal fisheries were aggregated from catch report cards and quality controlled by state agencies (including an estimated delayed mortality of 10% from catch-and-release). Non-tribal fisheries data included the catch of upstream stocks from refuge habitats, including Drano Lake (the mouth of the Little White Salmon River), the lower Wind River, the lower Deschutes River, and the John Day River Arm of John Day Reservoir [43]. Data for the tribal fisheries was collected from tribal creel sampling and expanded based on the number of fish tickets for commercial sale (primarily of Chinook salmon) to attempt to represent the total number of fishers.

The global survival model on the logit scale was:

$$\text{logit}(pSurv_i) = \beta_0 + s(\widehat{pDelay_i}) + \beta_1 O_i + \beta_2 A_i + \beta_3 J_i + \beta_4 H_j + \gamma_j, \qquad (4)$$

where $\beta_0$ is the intercept, $\beta_1$, $\beta_2$, and $\beta_3$ are parameters for the non-reference values of the categorical variables, $s(\widehat{pDelay})$ is a smoothing spline on $\widehat{pDelay}$ (knots again limited to 5), $H_j$ is the harvest rate in year *j* in which individual *i* migrated, and $\gamma_j$ is a random effect for return year *j*. As was done for the models of *pDelay*, model selection was performed by retaining the models with the lowest AICc values and we characterized goodness of fit using AUC.

## 2.7. Daily mortality rates

Due to the time it took for fish to pass the reach, survival of *fast* fish was assessed roughly a week post arrival at Bonneville Dam, on average, while survival of fish that delayed migration was generally assessed one to many months later, when those fish resumed active upstream migration as determined by their detection at McNary Dam. If a *fast* fish and a *slow* fish passed

Bonneville on the same day, the *fast* fish may have died upstream beyond the range of detection by the time the *slow* fish passed McNary Dam. We thus did not know how many *fast* fish were still alive on the same day that originally co-migrating *slow* fish were detected at McNary Dam. Some unknown portion of these *fast* fish were not alive at this point.

To account for the increase in mortality risk due simply to the duration of exposure in the reach, we estimated average mortality rates for fish (mortality per day) in addition to total survival. We achieved this by combining information from observed travel times and predictions from travel time models (Eq 2), migration behavior models (Eq 3), and our estimated probabilities of survival (Eq 4). First, travel times were simulated for fish that were not detected at McNary Dam (designated mortalities). For those fish, we simulated the category of migration behavior (*fast* or delaying) using the predicted probability of delay ($\widehat{pDelay}$) from Eq 3. If a fish was designated as delaying, *overwintering* (vs *slow* behavior) was simulated using a constant probability based on the total proportion of tags with this behavior in each population group (~0–3%).

While *fast* fish travel times were largely consistent across the migration period, *slow* and *overwintering* behaviors demonstrated strong seasonal patterns in the average duration of delays. To account for seasonal variation in travel times within behaviors, we simulated travel times ($\check{tt}_i$) using a normal distribution with the mean equal to the behavior-specific mean travel time prediction on the log scale (from Eq 2). We incorporated prediction variance around the mean as equal to the estimated variance in the predicted mean plus the estimated variance in travel times around the mean.

Daily mortality rates were calculated using the following equation:

$$MortR_i = (1 - pSurv_i)/tt_i^*,$$

(5)

where $pSurv_i$ is the predicted reach survival for fish $i$ from its respective survival model fit, and $tt_i^*$ represents either the observed travel time for fish $i$ or the simulated travel time ($\check{tt}_i$) if fish $i$ was not detected at McNary Dam. While an individual fish clearly cannot have a daily mortality rate, as it either lived or died over a specified time frame, this calculation aims to estimate a population-level daily mortality rate for fish that had the same traits and experienced the same migration conditions. We examined how *MortR* compared with the probability of delay, arrival date at Bonneville (*D*), and river temperature (*T*) using GAM smoothers fit to the set of estimated individual daily mortality rates.

## 3. Results

### 3.1. Arrival and migration behaviors

Fish from the Middle Columbia DPS generally arrived earliest at Bonneville Dam (median date = 29 July), followed by the Snake Early A-index (3 August), Upper Columbia DPS (9 August), Sal/Clear A-index (18 August), and finally the Sal/Clear B-index (13 September). This central migration period (July-September) was characterized by the seasonal peak of average river temperatures and declining flows in the study reach (Fig 3). Upper Columbia and Snake Early A-index steelhead encountered the highest average temperatures upon arrival (Median = 21.0°C, SD = 1.3°C), though all population groups encountered median temperatures upon arrival above 19.7°C (Table 1). Spill generally declined following peak temperatures in early August and the majority of Sal/Clear B-index fish arrived following the shutoff of spill operations on 1 September.

Mixture models (Eq 1) demonstrated that the probability of migration delay through the mid-Columbia varied across population groups. The Middle Columbia populations were most

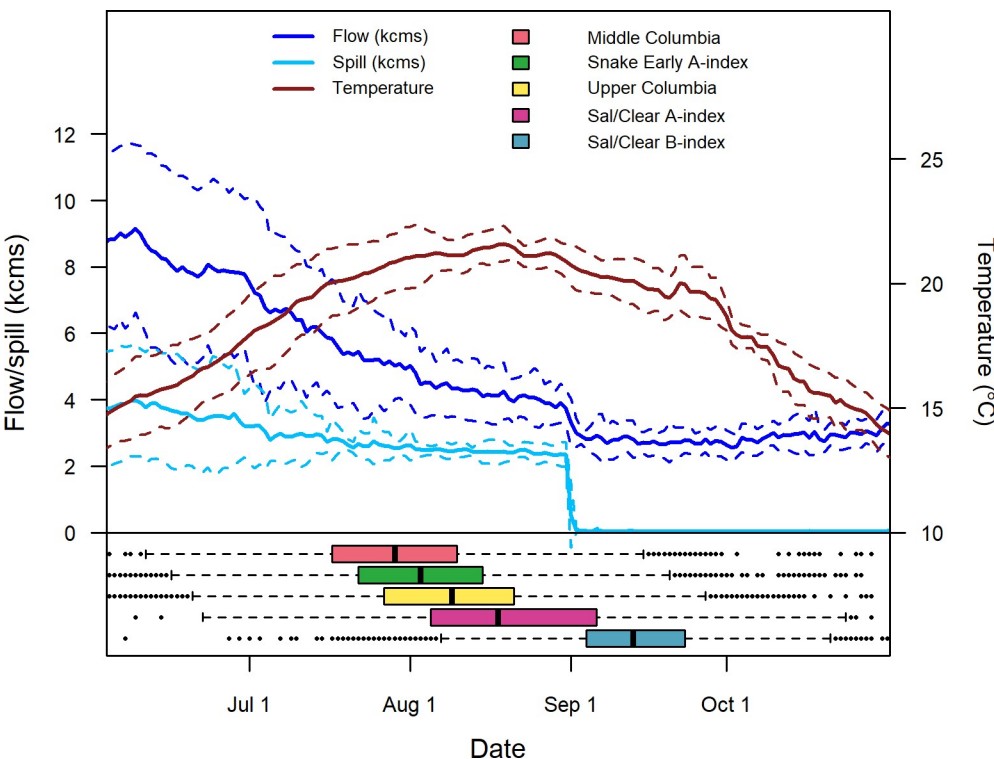

**Fig 3. Arrival date distributions by population group and the river environment.** Arrival dates for population groups (boxplots) plotted with mean daily values (solid lines) and associated bands for ± one standard deviation (dotted lines) for temperature, spill, and flow at Bonneville Dam for the study period 2004–2016.

likely to delay (*slow* or *overwintering* behaviors, 63%), followed by the Sal/Clear A-index (56%), the Snake Early A-index (54%), the Upper Columbia (43%), and the Sal/Clear B-index (42%) (Table 1, Fig 4A–4E). The majority of delayed individuals were categorized as *slow* migrants, with the estimated percent of *overwintering* fish ranging from a high of 3% in the Middle Columbia populations to being largely non-existent in the Upper Columbia

**Table 1. Median arrival conditions and predicted behavior distributions by population group.**

| | Middle Columbia | Snake Early A-index | Upper Columbia | Sal/Clear A-index | Sal/Clear B-index |
|---|---|---|---|---|---|
| | | **Arrival at Bonneville** | | | |
| Median date | July 29th | August 3rd | August 9th | August 18th | September 13th |
| Median temperature (˚C) | 20.8 | 21.0 | 21.0 | 20.7 | 19.7 |
| Median flow (kcms) | 4.4 | 4.2 | 4.1 | 3.6 | 2.8 |
| Median spill (kcms) | 1.9 | 1.8 | 1.8 | 1.5 | 0.02 |
| | | **Mixture model migration results** | | | |
| Proportion *fast* | 38% | 45% | 57% | 44% | 57% |
| Proportion *slow* | 60% | 53% | 43% | 55% | 41% |
| Proportion *overwinter* | 3% | 1% | 0% | 1% | 1% |
| Mean *fast* travel days | 8.6 | 8.6 | 7.8 | 8.7 | 7.4 |
| Mean *slow* travel days | 62.6 | 40.5 | 27.8 | 36.2 | 21.3 |
| Mean *overwinter* travel days | 234.0 | 233.5 | 195.4 | 191.8 | 179.2 |

Median conditions upon arrival at Bonneville Dam and mixture model results for migration behaviors.

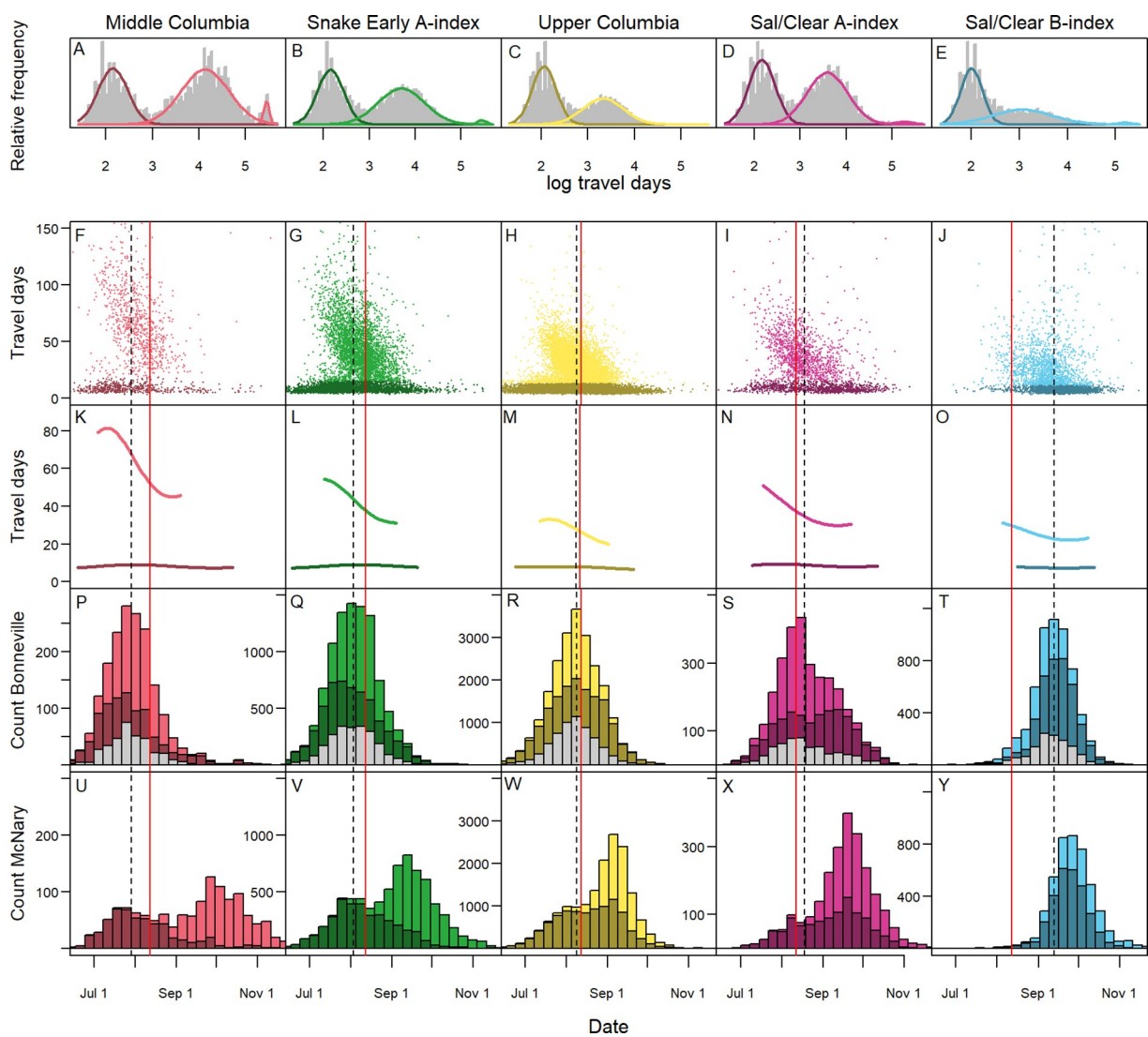

**Fig 4. Migration patterns by population groups.** Mixture model fits for each population group (A-E), scatterplots showing travel times by date for fish designated as *fast* (dark colors) and *slow* (light colors) respectively (F-J, *overwintering* fish shown in S1 Fig), GAMM smoothers for *fast* and *slow* fish respectively from Eq 2 shown for date ranges representing 95% of fish demonstrating respective behaviors (K-O), histograms of arrival dates at Bonneville Dam showing *fast* fish, delayed fish, and mortalities (in grey, P-T), and histograms of arrival dates at McNary Dam by migration behavior (U-Y). Vertical red lines represent date of peak temperatures averaged across study years and the vertical dashed lines represents median arrival date at Bonneville Dam for each population group (K-Y).

populations. Whereas estimated median travel times for *fast* migrants were fairly consistent across population groups (~7–9 days), travel times of *slow* migrants were quite variable. *Slow* migrating Middle Columbia fish generally took a couple of months (63 days on average) while Upper Columbia and Sal/Clear B-index fish were much faster (28 and 21 days respectively). Mean estimated migration travel times of *overwintering* fish ranged from a low of 179 days for the Upper Columbia to a high of 234 days for the Middle Columbia fish.

Smoothers for travel times by population group (Eq 2) demonstrated that while there was substantial variability in *slow* fish travel times during the migration season within and across populations (Fig 4F–4J), there were some shared seasonal patterns (Fig 4F–4O). *Slow* migrants tended to be slowest in July before peak temperatures in the beginning of August and quicker in late August and early September once peak temperatures had passed. Middle Columbia *slow*

fish tended to delay the longest while Upper Columbia *slow* fish delayed the least amount of time at a given date. In contrast, *fast* migrants had comparatively consistent travel times across the season (~7–10 days on average), though they took moderately longer during high temperatures. The earliest arriving *overwintering* fish tended to delay for the longest periods, overcompensating for their earlier arrival and thus leading to later detection at McNary Dam (Table 1 and S1 Fig). While arrival distributions were unimodal at Bonneville Dam (Fig 4P–4T), this combination of *slow*, *fast*, and *overwintering* migration behaviors results in arrival distributions being tri-modal for most population groups at McNary Dam (Fig 4U–4Y, note *overwintering* fish shown in S1 Fig).

### 3.2. Probability of delay

Covariate models for the probability of delay, *pDelay* (Eq 3), demonstrated consistent goodness-of-fits across population groups. AUC values were highest for the Sal/Clear A-index (0.81), followed by the Upper Columbia and Snake Early A-index (0.79), the Middle Columbia (0.78), and finally the Sal/Clear B-index (0.75). Temperature ($T$) was retained in all models and arrival date ($D$) was retained over the effects of flow ($F$) or spill ($S$) in each case (S3 Table).

The model results suggest that summer steelhead that encountered higher river temperatures at Bonneville Dam were more likely to delay, presumably to seek temperature refuge in the study reach (Fig 5A, uncertainty shown in S2A–S2E Fig). However the nature of the effect of temperature varied between the population groups with Middle Columbia and Sal/Clear

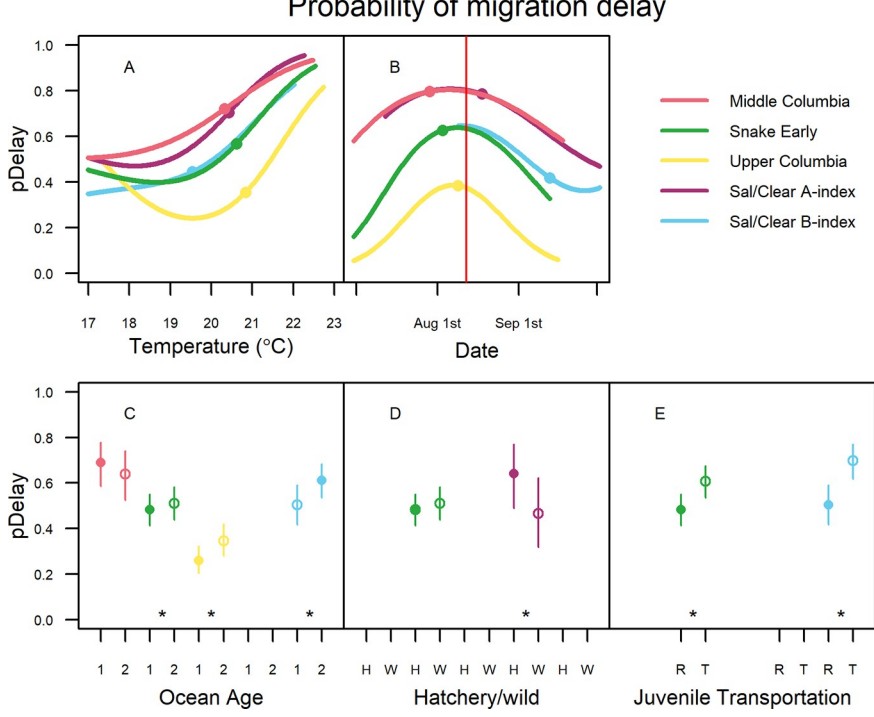

**Fig 5. Covariate effects for probability of delay models.** All variable effects are shown conditioned on the median or most common value of other variables in the entire dataset. All retained continuous variable effects contributed significantly ($P < 0.05$) and are shown for the 95% central extent of variable range for each population group with median values shown by points (A and B). Vertical red line represent date of peak temperatures averaged across study years (B). Significance of categorical variables (C-E) is shown by asterisks ($P < 0.05$) and uncertainty is shown by vertical lines. Filled circles in C-E indicate *pDelay* for the most common value of each categorical variable. *R* represents river-run (not transported) while *T* represents transported juveniles (E).

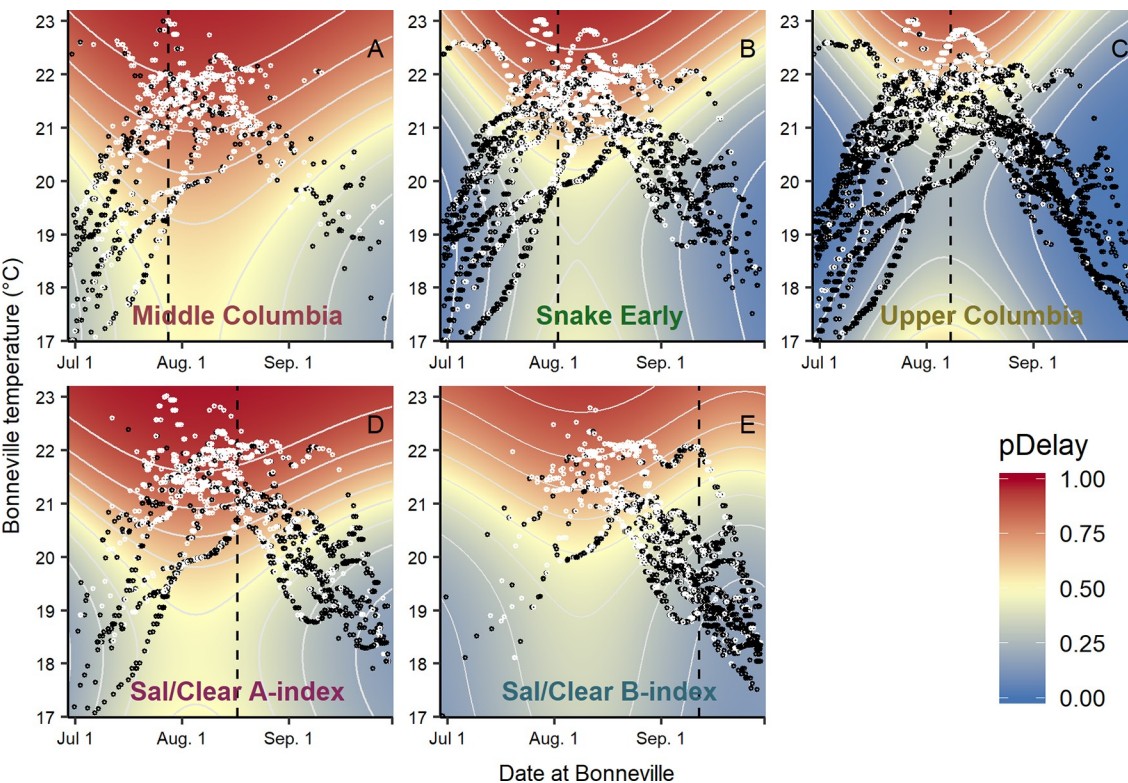

**Fig 6. Probability of delay surface.** The combined modeled effects for temperature and arrival date at Bonneville Dam for the probability of delay (*pDelay*) for each population group. Data used to inform models are plotted over the predicted surfaces with black circles representing designated *fast* migrants and white circles representing delayed migrants. Contour lines are spaced at probabilities of 0.1. Vertical dashed lines represent median arrival dates at Bonneville Dam by population groups.

A-index fish demonstrating the lowest temperature thresholds and Upper Columbia fish having the highest threshold. Additionally, temperature thresholds varied by arrival date at Bonneville Dam with all population groups demonstrating increased probabilities of delay at the end of July and beginning of August, just before river temperatures tended to peak (Fig 5B, uncertainty shown in S2F–S2J Fig).

Due to the seasonal relationship between temperature and date, it is useful to examine the regression surface produced by the combined effects of these two variables with the actual data used to inform the models plotted over the top. While few fish delayed in early July and September, the majority of fish in all population groups were predicted to delay their migrations during late July through August when stream temperatures tended to peak, with the exception of Upper Columbia fish (Fig 6). Due to the higher temperature threshold for delay estimated for the Upper Columbia, the majority of Upper Columbia fish were only predicted to delay their migration in late July through August during years that were warmer than average. Despite experiencing some of the highest temperatures during the study period, very few Upper Columbia migrants delayed their migration during anomalously warm temperatures in early July 2015 due to temperature thresholds being higher during these early migration dates (upper left corner of Fig 6C).

In addition to the effects of temperature and arrival date, models suggested that older fish were generally more likely to delay with the exception of the Middle Columbia and Sal/Clear A-index populations (Fig 5C). The effect of hatchery origin was generally not retained during model selection, though wild Snake Early A-index fish were less likely to delay (Fig 5D).

Finally, Snake River fish that were transported as juveniles were significantly more likely to delay than their run-of-river counterparts for Snake Early A-index ($P$ = 1.63e-15) and Sal/Clear B-index ($P$ < 2e-16) migrants (Fig 5E).

### 3.3. Behavior and survival

Estimated mean annual reach survival of tagged fish during the study period was highest for Sal/Clear A-index (82%) and B-index fish (81%), followed by the Middle Columbia (80%), the Snake Early A-index (77%) and finally Upper Columbia fish (76%, S2 Table). Selected covariate models for the probability of survival *pSurv* (Eq 4) demonstrated lower goodness-of-fits than those for the probability of delay, but were again fairly consistent across population groups. AUC values were highest for the Middle Columbia (0.61), followed by the Upper Columbia (0.60), the Sal/Clear A-index groups (0.59), Sal/Clear B-index (0.59), and the Snake Early A-index (0.56).

The predicted probability of delay $\widehat{pDelay}$ was retained with significant effects ($P$ < 0.05) in all selected models for the probability of survival (S4 Table). In all cases a higher probability of delay was associated with a reduced probability of survival (Fig 7A, uncertainty shown in S3A–S3E Fig). Given equal probabilities of delay, Upper Columbia fish had the lowest predicted probability of survival while Sal/Clear A-index fish had the highest. The populations groups that overall were least likely to delay (Sal/Clear B-index and Upper Columbia) demonstrated the steepest declines in survival with increased predicted probabilities of delay. Harvest rate was only retained for the later migrating Sal/Clear A- and B-index populations. When

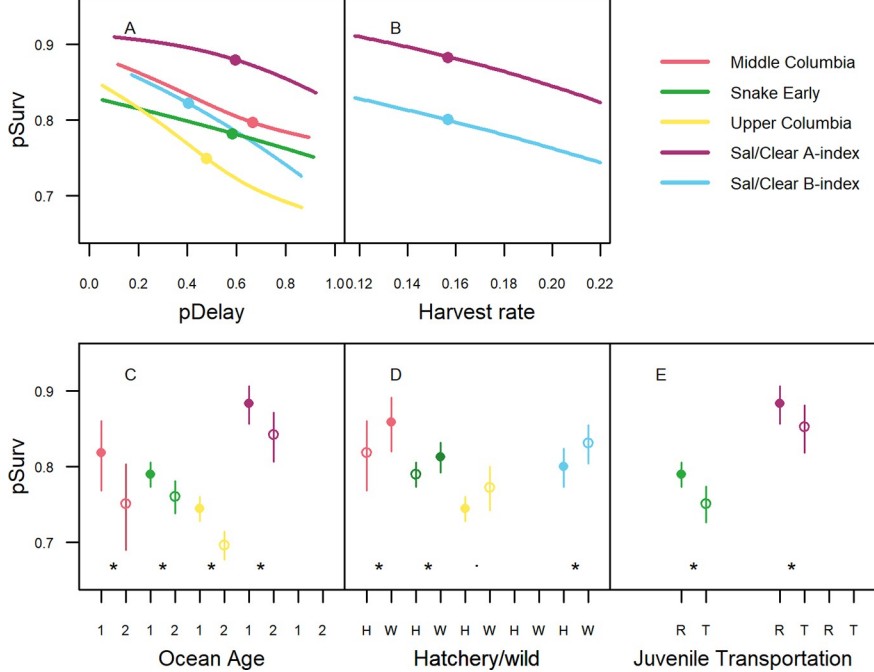

**Fig 7. Covariate effects for probability of survival models.** All variable effects are shown conditioned on the median or most common value of other variables in the model for the entire dataset. All retained continuous variable effects contributed significantly ($P$ < 0.05) and are presented for the 95% central extent of variables ranges by population groups with median values shown by points (A and B). Significance of categorical variables is shown by asterisks ($P$ < 0.05) and periods ($P$ < 0.1) and 95% confidence intervals are shown with vertical lines. Filled circles in C-E indicate the most common value of each categorical variable. *R* represents river-run (not transported) while *T* represents transported juveniles (E).

retained, higher harvest rates were associated with lower survival probabilities (Fig 7B, uncertainty shown in S3F–S3G Fig).

The effect of categorical variables in selected models for survival were largely consistent in direction and significant in all population groups. All selected models, with the exception of Sal/Clear B-index, found that older fish (ocean age 2) had lower survival than younger fish (Fig 7C), three out of five models retained effects showing lower survival of hatchery compared to wild origin fish (Fig 7D), and two out of three Snake River models found that fish transported as juveniles had reduced survival compared to their run-of-river counterparts (Fig 7E).

## 3.4. Daily mortality rates

While overall reach survival declined (i.e., reach mortality increased) when the probability of delay increased (Fig 7A), daily rates of mortality (*MortR*) generally decreased with increasing probability of delay (Fig 8A). This occurred because the proportional changes in travel times with delay were larger than the proportional changes in survival. However, in the population groups that less frequently exhibited delay, namely the Upper Columbia and the Sal/Clear B-index, there was an initial increase in daily mortality rates as the probabilities of delay increased to around 50% before a decline.

In general, daily mortality rates declined with higher river temperature (Fig 8B), suggesting that migration delay may have mitigated some of the impacts of high mainstem temperatures. However, we observed an initial increase in daily mortality rates with temperature before a decline at temperatures >20˚C in the Upper Columbia and Sal/Clear B-index population groups. All population groups demonstrated lower daily mortality rates during the late July/early August period during and just before peak river temperatures (Fig 8C). This aligned with the period in which the temperature thresholds for delay were the lowest and the predicted probabilities for delay were the highest (Fig 5B). Higher intensity of the fisheries in the late summer likely contributed to the rise in daily mortality rates during late August and September.

Upper Columbia fish again stood out as having the highest daily mortality rates while Middle Columbia and Sal/Clear A-index fish tended to have the lowest. Upper Columbia fish spent the shortest amount of time in the reach given the same arrival dates and river temperatures due to exhibiting the highest temperature thresholds for delay and relatively short delays

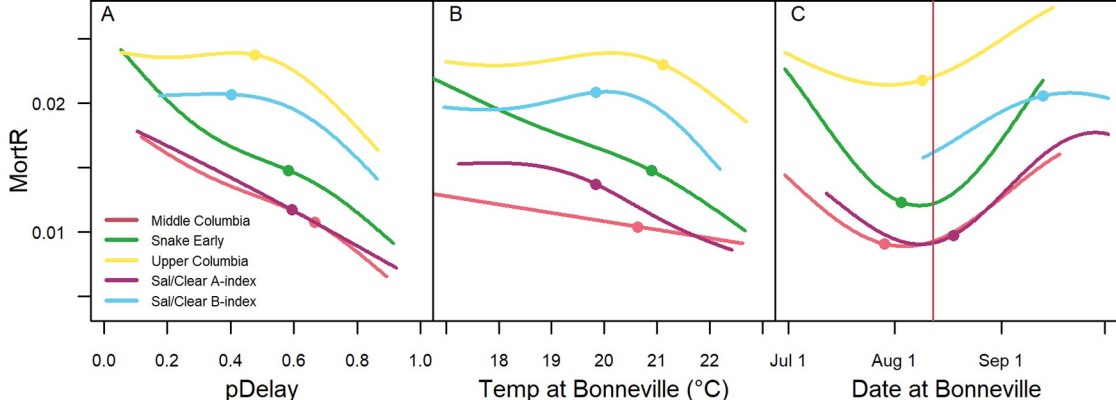

**Fig 8. Mortality rate relationships.** Modeled relationships between estimated daily mortality rates (*MortR*) and predicted probability of delay (*pDelay*) (A), temperature upon arrival at Bonneville Dam (B), and arrival date at Bonneville Dam (C). Median values by population group are shown by points. All lines were fit with GAM smoothers and demonstrated *P*-values < 0.005. Vertical red line represents date of peak temperatures averaged across study years (C).

when they did exhibit this behavior. Despite less temporal exposure in the reach, their overall survival was lower than their counterparts (76%) as a consequence of their high daily mortality rates. On the opposite end of the spectrum, Middle Columbia fish demonstrated the lowest temperature thresholds for delay, and their delays were the longest on average of all the population groups. Despite spending the most time in the reach, mean survival of Middle Columbia fish was comparatively high (80%) due to their low daily mortality rates.

### 3.5. Interannual variability in delay and survival

The majority of fish, except the late arriving Sal/Clear B-index, arrived in the study reach from 15 July-1 September (64%-78%, Sal/Clear B-index = 18%), when river temperatures were highest and temperature thresholds for delay were lowest. We examined how the annual proportion of fish that delayed migration and that survived varied with mean river temperature within this date range (Fig 9A). The majority of fish that arrived during this period delayed their migration except in the Upper Columbia group, where the majority only delayed during the hottest years (Fig 9C and 9D). A significantly higher proportion of fish delayed ($P = 0.09$ for Middle Columbia and $P < 0.05$ for all others) during this period in years with higher mean river temperatures (Fig 9D), which ranged from approximately 19.5°C to 22°C. The proportion of fish that migrated slowly during this period was positively correlated across population groups (Fig 9C, Pearson's $r = 0.48$ to 0.81) due to their shared response to temperature. However, survival was less correlated across years between population groups ($r = -0.21$ to 0.49), but synchrony increased following 2010 ($r = 0.29$ to 0.73, Fig 9E). Despite fish spending more time in the reach during the peak migration period during warm years, relationships between mean temperature and survival were weak ($P > 0.l$; Fig 9F), with the exception of the Upper Columbia ($P = 0.08$). Similarly, we found that the mean proportion of fish that delayed was only significantly ($P = 0.001$) related to the mean proportion that survived in Upper Columbia steelhead (Fig 9B). In summary, on an annual basis temperature was strongly related to migration behavior for all summer-run steelhead population groups, but it was not strongly related to survival in any group with the exception of Upper Columbia steelhead.

## 4. Discussion

### 4.1. Migration behavior

Our results demonstrated shared patterns in migration behavior across inland populations of Columbia River summer-run steelhead, while highlighting substantial differences. All population groups demonstrated analogous seasonal patterns in the probability of delay with stream temperature as the primary trigger, but with lower temperature thresholds for delay just before and during peak river temperatures. Additionally, the average length of delays tended to be substantially longer in July before peak stream temperatures than in August during or after peak temperatures. However, in alignment with our hypothesis, the temperatures at which fish became more likely to delay varied across population groups and the length of average delays ranged from around three weeks to over two months.

Lower temperature thresholds for delay and longer delays just before the peak of river temperatures may be an adaptive response as delaying migration at this point allows fish to avoid actively migrating during the hottest river temperatures. Migrating during high temperatures is energetically expensive and may impact the ability of migrants to finish their migration or to survive the duration of freshwater holding to the spawning period. Additionally, temperatures over 21°C, which summer-run steelhead commonly encounter in this reach, can be physiologically harmful to steelhead [22].

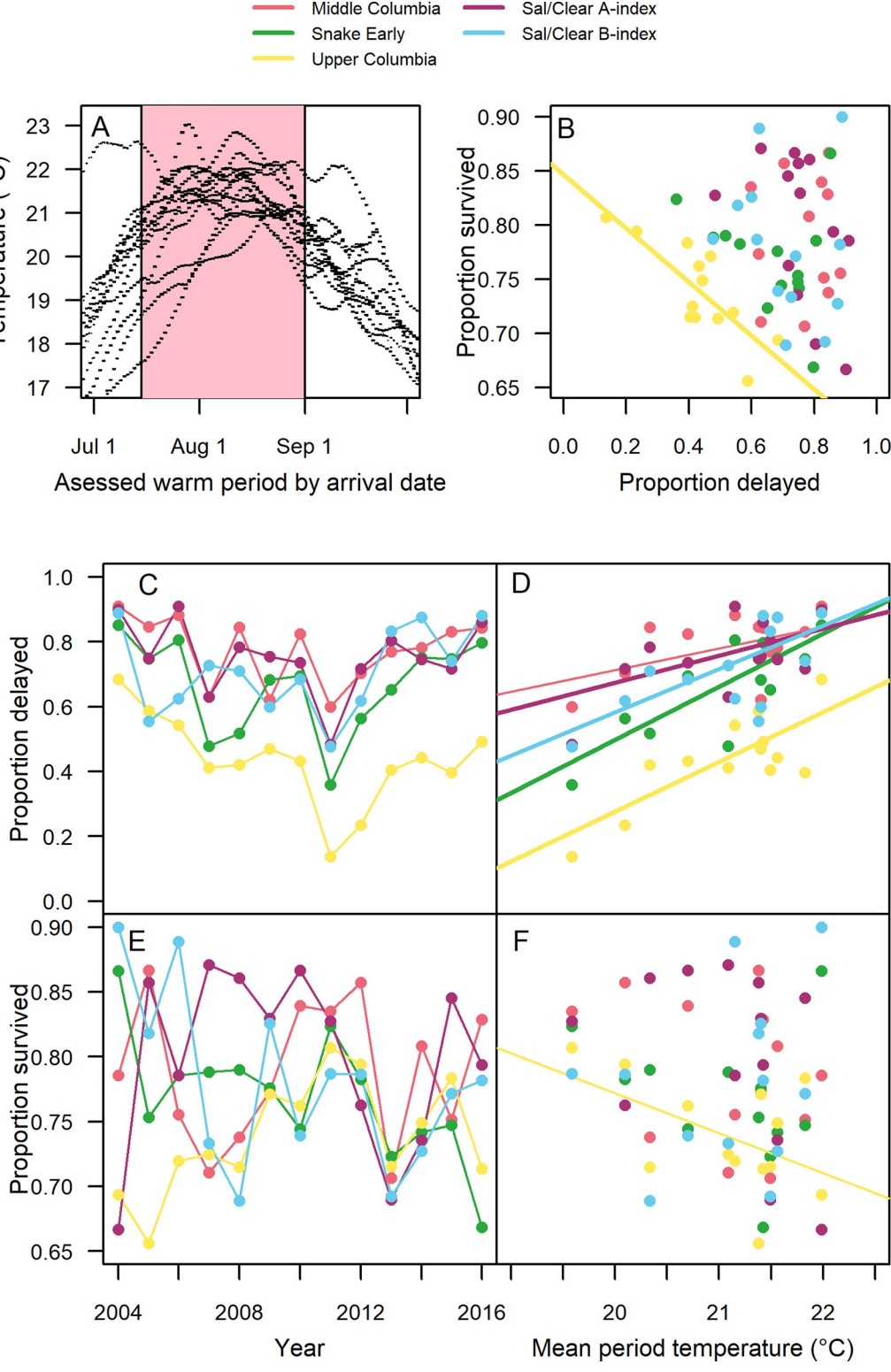

**Fig 9. Interannual variability in behavior and survival.** Interannual variability in the proportion delayed and survival by year assessed for the July 15th-Septeber 1st warm period when fish were most likely to delay migration (A). Panel B shows a direct comparison between the proportion delayed and the proportion that survived (B). The mean proportion of migrants that delayed (C) and the proportion that survived the reach (E) are shown by year and compared to the average reach temperature (D and F) for each population group. Significant linear model relationships are shown with thick ($P < 0.05$) or thin ($P < 0.1$ & $> 0.05$) solid lines (B, D and F).

However, observed migration delays during this high temperature period may to some extent be an involuntary response of fish due to migration impediments at dams. Altered environmental conditions at dams, such as high dissolved gas concentrations due to spill [44] and temperature gradients in fish ladders [45], can stress fish and delay dam passage. While some fish may simply take extra time to pass a dam under such conditions, others may be forced to seek temperature refuge to recover following stressful dam passage failures or ladder ascensions.

## 4.2. Behavior and survival

Our results suggest that migration delay is associated with lower reach survival. However, direct comparisons of reach survival between *slow/overwintering* migrants and *fast* migrants only describe a partial picture due to the differences in travel times between the behaviors. Survival for *fast* migrants is measured roughly eight days following arrival at Bonneville Dam (the mean travel time to McNary Dam) while survival of fish that delay is measured weeks to months later. Accordingly, when the survival of a delayed fish is counted at McNary Dam, we do not know if the *fast* migrants that entered the reach at the same time are still alive further upstream. Similarly, we do not know whether mortality events occurred within 8 days (median *fast* travel time) for fish that were likely to delay.

In alignment with our results, previous research also found that migration delays of summer-run steelhead were related to lower survival through the study reach, suggesting that mortality was likely higher as a consequence of increased fisheries exposure [27]. However, the reproductive success of summer steelhead depends not only on their ability to complete their migration to spawning tributaries, but also on their ability to survive in freshwater for many months before they spawn. *Fast* migrants may have higher migration survival due to spending less time in the reach, but we do not know if this translates to higher survival until the spring spawning season because they still must hold somewhere else in the watershed until spawning.

We found that migration delay was associated with lower daily mortality rates, despite producing higher overall reach mortality. In short, this means that the increase in mean travel times in response to high river temperatures was generally proportionally larger than the associated increase in reach mortality. In addition, we found that even though high proportions of fish delayed migration during warm years, there was no strong relationships between mean river temperatures and annual reach survival, with the exception of the Upper Columbia which demonstrated the highest temperature threshold for delay. These mixed results could be in-part a consequence of actively migrating fish being more exposed to gillnetting and dip netting fishing techniques, common in the tribal fishery. Alternatively, the strategy of migration delay to seek temperature refuge may help mitigate migration mortality in the reach due to the avoidance of high river temperatures. This behavioral flexibility, which is not exhibited by other Columbia River salmon species to nearly the same extent [25], may help steelhead respond to anticipated increases in river temperatures with climate change assuming that temperature refuge habitats continue to be accessible.

We note that there are in-stream PIT tag detection arrays in many tributaries that could provide further insight into the consequences of different behavioral decisions on migration survival. However, these arrays have low and variable detection efficiencies compared to the dams and some initial explorations have suggested that efficiencies may depend on river flows. Furthermore, steelhead behavior is similarly complicated upstream from the study reach. Fully accounting for these complex movements given the lower detection efficiencies requires large sample sizes, which to date only exist for fish PIT-tagged at Rock Island Dam in the Upper Colombia [46]. Consequently, effectively characterizing these complexities was beyond the scope of this manuscript.

Accordingly, it is important to keep in mind that the impacts we describe on migration survival through the study reach represent only one aspect of the total fitness consequences of alternative migration strategies. Stressful conditions and migration delays lead to higher energetic expenditures for fish [47], which may leave fish more susceptible to disease, predation, and other sources of mortality. Energetically costly migrations in salmonids are associated with increased pre-spawn mortality for fish that survive their migration to reach spawning grounds [48, 49], though less is written on this subject for steelhead specifically (e.g., [50]). In addition, stressful migrations can lead to reduced development of gametes, diminished aggressiveness and longevity in competition for mates and prime redd locations, and lower egg incubation survival [51, 52], which may combine to reduce the reproductive fitness of successful spawners. Future studies would need to assess impacts on reproductive success to fully account for the consequences of different migration behaviors. This could potentially be achieved using genetically based parentage analysis, which is already used extensively in the Colombia River Basin (e.g., [53, 54]).

## 4.3. Interpreting categorical effects

Our models estimated consistently lower survival across populations for older individuals (~2–7% relative difference) and for hatchery fish (~3–4%) compared to wild fish in three of the population groups. These survival differentials may be a consequence of harvest in the reach. While the tribal fishery does capture wild-origin fish, the number of wild fish caught is limited and the regulations of the recreational fisheries in the reach only allow take of hatchery fish. Mean exploitation of hatchery fish in the reach (Zone 6) during the 2004–2016 study period (~15.2%) was estimated to be significantly higher than wild fish (8.7%; data provided by Jeromy Jording, NOAA Fisheries). These harvest rates represent a large proportion of the estimated average total mortality in the study reach. Additionally, even if fish are not harvested, encounters with fishing gear may stress or injure fish, contributing to pre-spawn mortality or reduced spawning success (e.g., [50]), and these impacts are exacerbated at higher temperature [55].

Hatchery fish may also have lower survival because of a general tendency toward reduced fitness. Lower fitness of hatchery fish compared to their wild counterparts is common in salmon [56] and may be a consequence of relaxed selectivity on adaptive traits or maladaptive selection in the hatchery rearing process [57–59]. The Upper Columbia populations had a very high proportion of hatchery origin fish in our analysis (95%), which may be related to their relatively lower survival and apparent diminished ability to respond to high temperatures.

Lower survival of older steelhead, which tend to be larger, may be a consequence of size selectivity in the fishery (e.g., [60]), though a higher metabolic cost of high temperatures for larger fish could also be a factor [61]. There is some evidence that fisheries within the reach disproportionately harvest older individuals [62, 63]. A large part of the tribal fishery uses gillnets to target fall Chinook salmon (*Oncorhynchus tshawytscha*), catching steelhead as bycatch. Gillnet mesh sizes targeting Chinook are likely to disproportionately capture larger steelhead since Chinook are larger on average. Selective terminal fisheries have the potential to reduce the size and age distributions of salmon populations [64] and at least Dworshak hatchery fish have demonstrated declines in age and size consistent with such selection [63]. This possibility should be further explored as the loss of the oldest and largest individuals could impact population productivity and the viability of fisheries [65, 66]. Larger females generally have higher fecundity [67], contributing disproportionately to the next generation [66], and can dig deeper redds which are more resilient to scour and desiccation [68].

While we believe that higher survival of younger and wild individuals was at least partly a consequence of the fisheries, our simple metric of annual fisheries exploitation only improved

models for the late migrating Salmon and Clearwater River populations, but not the other population groups. Our finding of a larger impact of the harvest rates on Salmon and Clearwater fish is likely a consequence of seasonal exploitation patterns. The fall Chinook run overlaps with the later part of the steelhead run and thus exploitation of steelhead is thought to be higher for later migrating fish. During this period the run is disproportionately composed of Salmon and Clearwater steelhead, which likely leads to higher exploitation rates compared to earlier arriving populations. However, given that estimated exploitation rates accounted for a large proportion of mortality in this reach, combined with the estimated effects of ocean age and rear type origin, it seems unlikely that the fisheries were inconsequential to earlier-migrating populations. This result highlights the need to better estimate population-specific harvest within the season to improve our understanding of the impacts of the fisheries on specific populations, which are likely uneven.

In addition, our models estimated negative effects of downstream juvenile transportation on adult migration success in Snake Early and Sal/Clear A-index populations (~3–4%). Carry-over impacts of transportation on later life-history stages have previously been documented for steelhead, as well as for other salmon species [69, 70]. Negative impacts from transportation have been associated with increased rates of straying to non-natal tributaries and non-linear migration movements [71–73], though straying rates appear higher in Chinook compared to steelhead [74]. Increased straying of transported fish is likely a consequence of reduced and interrupted olfactory imprinting as juveniles [75–77]. This result provides further evidence that the gains in downstream juvenile survival provided by barging should be weighed against the delayed impacts on subsequent life stages to maximize the benefit of this management strategy [78].

### 4.4. Understanding population differences

As we hypothesized, we documented substantial differences in thresholds for and lengths of migration delays between population groups. Upper Columbia steelhead stood out as the population group that was least likely to delay migration on a given date, demonstrating the highest temperature threshold for this behavior. Additionally, they generally exhibited the shortest migration interruptions when they did delay at a given date/temperature. As a consequence of their resistance to delay, Upper Columbia fish appeared to suffer lower survival through the reach during hot years compared to other populations. Sal/Clear B-index fish also demonstrated low rates of migration delay, but this appeared to be more a consequence of their late arrival instead of higher temperature thresholds. On the other end of the spectrum, Middle Columbia fish had the lowest temperature thresholds for delay and delayed for the longest time on average.

Population differences in migration behavior may in-part be due to genetically derived distinct behavioral thresholds evolved to account for the distinct migration routes and conditions within respective spawning tributaries. For example, Middle Columbia tributaries (e.g., Walla Walla and Yakama rivers) tend to be very warm in their lower reaches during the summer, which may create migration barriers to fish attempting to access cold-water holding habitats in upper reaches. Accordingly, it may be more beneficial, or even a necessity, for Middle Columbia migrants to find cold-water holding habitats elsewhere until temperatures in natal streams drop in the fall. Middle Columbia fish have also been shown to overshoot their tributaries and hold in the Snake River or Upper Columbia before returning to natal streams to spawn [10]. In contrast, Upper Columbia populations originate in more mountainous rivers which contain cooler temperatures in lower reaches and the Upper Columbia River itself remains cooler than the mid-Columbia and Snake River. This likely reduces the advantage of

these fish delaying in downstream tributaries unless mainstem migration conditions become especially detrimental.

However, it is also possible that Upper Columbia steelhead have lost behavioral traits that allow them to functionally respond to high temperatures due to hatchery introgression and low levels of remaining genetic diversity [18]. These factors likely contribute to low replacement rates in these populations [18]. In contrast, Middle Columbia fish, which were the most likely to delay, had the lowest percentage of hatchery fish in our database (41%). Hatchery introgression is relatively high in other populations and the impacts of lost genetic diversity on behavior should be further considered.

While migration delays in the study reach have previously been associated with higher mortality due to extended exposure to the fisheries [27], contrary to our expectations we found that mortality was fairly consistent on average across our population groups (annual average ranged from ~23% to 18%) despite substantial differences in thresholds for delays and average travel times. In fact, despite spending relatively little time in the study reach (mean travel time of ~16 days), Upper Columbia steelhead demonstrated the lowest average estimated survival of the population groups. In contrast, Middle Columbia fish generally spent the most time in the reach (~55 days) while exhibiting relatively high estimated survival. In alignment with this result, we found no significant relationship between the annual proportion of fish that survived the reach and the proportion of fish that delayed or mean river temperature in any population group except the Upper Columbia. These results suggest that there were other factors besides river temperature and the resulting migration behavioral decisions that drove interannual variability in survival for most populations.

There are likely behavioral differences across populations that we cannot fully account for here that may impact survival. For example, given the large variation in the duration of migration delays, there are bound to be population-specific variation in the habitat that fish use for temperature refuge. Fish may either delay in tributary confluence water plumes in the mainstem Columbia or enter tributary rivers [27, 79], and these habitats may provide unequal benefits as temperature refuges. In addition, differences in habitat use may interact with spatial and temporal heterogeneity in the fisheries, which likely contributes to uneven patterns in exploitation across populations. For example, the state of Oregon recently imposed steelhead fishery closures inside thermal refuge sites while the state of Washington did not.

## 5. Conclusions

As river temperatures continue to increase in the Columbia River with climate change [70, 80–82], our models suggest that summer-run steelhead will delay their migrations more frequently to seek out the cold-water refuge habitats of the Columbia River Gorge area. Given the extensive geographical distribution of steelhead and their exposure to climate impacts (3), it is likely that populations from other watershed will demonstrate similar responses during the adult migration. This behavioral flexibility may allow steelhead to better respond to climate impacts compared to other salmon species that do not exhibit this trait. In addition, steelhead are able to utilize refuge habitats during juvenile rearing to support growth and consistent outmigration timing (e.g., [83]). However, the benefits of refuge habitats depends on them remaining accessible and cool, despite potential future impacts from continued climate change and landscape changes. It is therefore essential to identify, protect, and restore these important habitats in the Columbia River and in other basins to ensure that they continue to provide resiliency to the salmon and steelhead populations that depend on them [83, 84].

Increased use of refuge habitat with climate change will also increase the temporal exposure of Columbia River summer-run steelhead to the fisheries in the reach, which will be an

important consideration for management given that estimated exploitation rates accounted for a high percentage of estimated mortality. Managers should ensure that future harvests do not become excessive. Our inability to describe a fisheries impact in three out of five of our population groups, despite high overall estimated exploitation rates, suggests that the impact of the fisheries is uneven and that assessments of mortality in the reach would benefit from better stock-specific estimates of exploitation. The information provided here could help fisheries managers better account for stock-specific fisheries exposure to avoid the over-exploitation of protected populations. Interpretation of these results would be improved with more detailed information on fish movements during migration delays, including additional studies on the spatial and temporal patterns of habitat use (e.g., radio tagging [25, 27]).

It is important to reiterate that our analysis covers only one reach during the adult migration life history stage. Accordingly, our results should be interpreted in this context. We focused on the mid-Columbia reach because this is where fish most commonly seek temperature refuge [25], where fishing pressure is high, and because it is a shared migration reach for inland populations. However, fish seek temperature refuge in other reaches [25, 85], overwintering appears to be more common for most populations in reaches further upstream [10], and delayed impacts from stressful migrations on reproductive success are likely. Future studies tracking individual populations from their migration through reproduction could better illuminate the full extent of the tradeoffs between different migration behaviors.

## Supporting information

**S1 Table. Sample sizes by steelhead population groups, specific populations, and return years.** For fish tagged and released in the Clearwater River, the fish from the South Fork Clearwater and Middle Fork Clearwater, as well as those from Dworshak National Fish Hatchery and Lolo Creek, were considered B-index. For the Salmon River, fish from the Middle Fork Salmon and South Fork Salmon were considered B-index. Additionally, both A-index and B-index hatchery fish have been released at the Pahsimeroi River trap in recent years. Hatchery fish at this trap were identified as either A-index or B-index based off of their stated stock name in the PTAGIS database. All other fish from the Salmon and Clearwater rivers were included in the Salmon/Clearwater A-index migration group.
(DOCX)

**S2 Table. Summary of run statistics by steelhead population groups and year.** Shown are survival from Bonneville Dam-McNary Dam, mean travel time from Bonneville Dam-McNary Dam, median day of arrival at Bonneville Dam, proportion hatchery fish (vs. wild), mean ocean age, and proportion of fish that were transported.
(DOCX)

**S3 Table. AICc model selection table for the probability migration delay (*pDelay*).** Variables considered included ocean age (*A*), reartype origin (*O*), juvenile transportation history (*J*), smoothers for river temperature s(*T*), river flow (*F*), dam spill (*S*), and arrival date at Bonneville Dam s(*D*), and a random effect for year *y*. Models including one or none of the smoothers s(*D*), s(*F*), s(*S*) were compared. Coefficient values given for intercept. Number of parameters represented by *np*.
(DOCX)

**S4 Table. Model selection for survival models.** AICc model selection table for the probability of survival (*pSurv*). Variables considered included ocean age (*A*), reartype origin (*O*), juvenile transportation history (*J*), a linear effect for annual harvest (*H*), a smoother for the predicted probability of delay s($\widehat{pDelay}$), and a random effect for year *y*. Coefficient values given for

intercept and linear effect of *H*. Number of parameters represented by *np*.
(DOCX)

**S1 Fig. Travel times and arrival dates for overwintering steelhead.** Travel times and arrival dates at Bonneville Dam and McNary Dam for *overwintering* fish. Smoothed relationships between arrival dates and travel times at each dam are shown.
(TIFF)

**S2 Fig. Fitted continuous variable effects for the probability of delay models with confidence intervals.** Fitted effects for selected continuous variables (*Temperature* and *Date*) from probability of survival models showing 95% confidence intervals.
(TIFF)

**S3 Fig. Fitted continuous variable effects for the probability of survival models with confidence intervals.** Fitted effects for selected continuous variables (*pDelay* and *Harvest Rate*) from probability of survival models showing 95% confidence intervals.
(TIFF)

**S1 Data.**
(ZIP)

**S2 Data.**
(ZIP)

**S1 Script.**
(R)

## Acknowledgments

Thanks to Jeromy Jording for providing harvest estimates. Thanks to Jeromy Jording and Thomas Buehrens for discussing how the fisheries are implemented, managed, and monitored. Thanks to Aimee Fullerton, Chris Tatara, Kevin See, Ben Sandford, and Janet Yood for providing constructive comments. Thanks to Jim Faulkner for comments, statistical guidance and an internal review and Kevin See for an internal review. Thanks to representatives from the Idaho Department of Fish and Game and the Columbia River Inter-Tribal Fish Commission for comments on early versions of this analysis. Thanks to one anonymous reviewer and the Matt Keefer for constructive comments during peer review. Finally, the authors would like to thank the numerous state, federal, and tribal agencies whose extensive tagging and monitoring efforts contributed to the PTAGIS database.

## Author Contributions

**Conceptualization:** Jared E. Siegel, Lisa G. Crozier.

**Data curation:** Jared E. Siegel, Lauren E. Wiesebron, Daniel L. Widener.

**Formal analysis:** Jared E. Siegel, Lauren E. Wiesebron.

**Investigation:** Jared E. Siegel.

**Methodology:** Jared E. Siegel, Lisa G. Crozier, Lauren E. Wiesebron.

**Project administration:** Lisa G. Crozier.

**Supervision:** Lisa G. Crozier.

**Validation:** Jared E. Siegel.

**Visualization:** Jared E. Siegel.

**Writing – original draft:** Jared E. Siegel.

**Writing – review & editing:** Jared E. Siegel, Lisa G. Crozier, Lauren E. Wiesebron, Daniel L. Widener.

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
