## [Decision Letter · Decision Letter 0]

10 Dec 2020

PONE-D-20-33797

Environmentally triggered shifts in steelhead migration behavior in the middle Columbia River

PLOS ONE

Dear Dr. Siegel,

Thank you for submitting your manuscript to PLOS ONE. After careful consideration, we feel that it has merit but does not fully meet PLOS ONE’s publication criteria as it currently stands. Therefore, we invite you to submit a revised version of the manuscript that addresses the points raised during the review process.

Two reviews offer minor albeit important points to consider and edit - these are mandatory.  In general, the manuscript is well written and worthy of publication. These minor changes should not be too difficult for the authors to complete. 

We look forward to receiving your revised manuscript.

Kind regards,

Madison Powell, PhD

Academic Editor

PLOS ONE

Journal Requirements:

2. Our internal editors have looked over your manuscript and determined that it is within the scope of our Freshwater Ecosystems Call for Papers. This collection of papers is headed by a team of Guest Editors for PLOS ONE (https://collections.plos.org/s/freshwater-ecosystems). The Collection will encompass a diverse range of research articles on biodiversity conservation, including freshwater fish ecology. Additional information can be found on our announcement page: https://collections.plos.org/s/freshwater-ecosystems.

If you would like your manuscript to be considered for this collection, please let us know in your cover letter and we will ensure that your paper is treated as if you were responding to this call. If you would prefer to remove your manuscript from collection consideration, please specify this in the cover letter.

3.We note that [Figure(s) 1] in your submission contain map images which may be copyrighted. All PLOS content is published under the Creative Commons Attribution License (CC BY 4.0), which means that the manuscript, images, and Supporting Information files will be freely available online, and any third party is permitted to access, download, copy, distribute, and use these materials in any way, even commercially, with proper attribution. For these reasons, we cannot publish previously copyrighted maps or satellite images created using proprietary data, such as Google software (Google Maps, Street View, and Earth). For more information, see our copyright guidelines: http://journals.plos.org/plosone/s/licenses-and-copyright.

1.    You may seek permission from the original copyright holder of Figure(s) [1] to publish the content specifically under the CC BY 4.0 license. 

4. We note you have included a table to which you do not refer in the text of your manuscript. Please ensure that you refer to Table 2 in your text; if accepted, production will need this reference to link the reader to the Table.

Additional Editor Comments (if provided):

Two reviews offer minor albeit important points to consider and edit. In general, the manuscript is well written and worthy of publication. These minor changes should not be too difficult for the authors to complete.

Reviewers' comments:

Reviewer's Responses to Questions

**Comments to the Author**

1. Is the manuscript technically sound, and do the data support the conclusions?

Reviewer #1: Yes

Reviewer #2: Yes

2. Has the statistical analysis been performed appropriately and rigorously? 

Reviewer #1: Yes

Reviewer #2: Yes

3. Have the authors made all data underlying the findings in their manuscript fully available?

Reviewer #1: Yes

Reviewer #2: Yes

4. Is the manuscript presented in an intelligible fashion and written in standard English?

Reviewer #1: Yes

Reviewer #2: Yes

5. Review Comments to the Author

Reviewer #1: This manuscript examines the variation in migration behavior of summer-run steelhead (anadromous rainbow trout, Oncorhynchus mykiss) through the Bonneville-McNary reach using PIT tag data from fish tagged as juveniles from different populations in the middle upper Columbia River basin, associated estimates of mortality, and the association of various environmental factors. Steelhead that migrate through this reach, which exhibits higher water temperatures due to retention time of the water in this lower reach, often seek thermal refugia and delay their migration to upper reaches for spawning. The authors find that populations show both similarities, such as a delay in migration that proceeded the highest annual temperatures and deceased daily mortality risk with delay despite increased cumulative within-reach mortality, and differences, including the temperature threshold at which delay is made, rates of delay, and associated mortality rates. As steelhead in the middle and upper Columbia are with few exceptions ESA-listed, and understanding the natural and anthropogenic factors that influence mortality in this therefore of great interest, I find that this paper is timely, adequately executed, and an insightful contribution to the relevant literature.

I would recommend, at the authors’ discretion, that they consider clarifying their manuscript in one respect, which the authors allude to, but is not in my opinion sufficiently identified. The ultimate question for migrating Columbia River steelhead is one of total fitness (survivorship to spawn, fecundity, and survivorship of their offspring to reproduction). While migration through the lower Columbia reaches and the decision to delay and seek thermal refugia may create some differential in survivorship out of that reach, the survivorship of fish that do not delay but make it through the reach is not secured, as the authors mention, and more importantly even if they do survive to spawning tributary, this does not necessarily guarantee greater total fitness, since the additional components of fitness may also be affected by the choice to continue migrating through adverse conditions. Again, as the authors allude, a better measure of the cost/benefit (“bet hedging” advantage) provided by migration delay would be to measure the actual reproductive success of fish that delayed or continued through, which would be a cumulative result of those choices. While this is clearly beyond the scope of this study, I think it’s important to clarify that the patterns identified herein only represent one aspect of that overall measure.

One other minor concern that I would have wished the authors to have addressed was that it was odd to me that the authors identified three distinct patterns in the data, both conceptually and in the fit of tri-modal models to the migration timing data. I have always inferred the “decisions” to 1) delay migration and seek thermal refugia and 2) overwinter outside of spawning tributary to be distinct, since the first (presumably) implies an active choice to seek a thermal refuge (usually a non-natal tributary), while the latter is likely more often simply pausing active migration because of lower kinetic limits. Moreover, the authors hardly discuss overwintering as a strategy distinct from the “slow” form of delay in the results, and indeed include them as a joint probability in additional models, which I think would rather make sense (in my mind, they are not distinct vis-à-vis initial choices to delay or not). Moreover, just because a fish does not overwinter in this lower Columbia stretch does not mean that it did not overwinter farther up but still outside the spawning tributary, a facet of fitness that is not assessable here. Given this, I wondered why it was necessary to consider these distinct modes at all. Relatedly, I wondered, given that despite obvious modes in the raw and log-transformed run-timing data there is considerable overlap in the two distributions associated with the ‘fast’ and ‘slow’ strategies, what the effect of arbitrarily assigning fish to either category based on highest probability (line 201). I wondered if there was a model variation that would allow using assignment probabilities directly rather than a priori assignment, or why the authors did not explore/utilize that.

Reviewer #2: General comments.

This was an interesting, well executed study and a well-written manuscript. Overall, I found the results convincing and most of the conclusions appeared to be reasonably well defended. I have three broadly general concerns and a number of mostly minor questions and suggestions (see specific comments below). My first concern regards why the authors chose to ignore the extensive additional PIT-tag detection data that were collected for the studied steelhead upstream from McNary Dam? Most readers familiar with the Columbia River basin will recognize that these additional data were available and could have provided considerably more information about the survival questions addressed in the study. My second concern is that the authors have identified a daily-scale survival benefit from cool and cold water refuges, but they have provided little to no commentary about the need to protect or restore these habitats in light of projected regional climate warming. Ensuring the persistence of such habitats is central to ongoing management efforts in the basin. Third, I think the manuscript would be more effective and potentially reach a broader audience if some effort was made to broaden the geographical scope of the messaging. Steelhead are widely distributed and many populations are vulnerable to warming river conditions, especially in the southern portion of the species’ range.

I appreciate the opportunity to review this work and to see how the authors have built upon some of the early studies on this subject by our research lab.

Sincerely,

Matthew Keefer, University of Idaho

Specific comments.

1. Title. The current title captures the behavior element of the study but neither the reach survival nor instantaneous mortality aspects.

2. Animal Research. Although the study authors did not handle any fish in this study, approval was presumably required for the dozens of PIT-tagging studies that made this analysis possible. Please also see next comment.

3. Data Availability. The PIT-tag data used in this study were available only due to years of effort by many state, federal, and tribal agencies. The PTAGIS data-use-policy states:“Data users should contact data contributors to gain context and to ensure that their intended use of the data is appropriate.” It is not clear to me in the disclosures that the authors took this step, though perhaps there is a special arrangement between NOAA-Fisheries and the juvenile steelhead tagging groups? Regardless, I think the author should – at a minimum - include a nod to the participating agencies in the Acknowledgments section and give some lip service to the appropriateness of included data.

Line 25. ‘tag data = ‘tag detection data’

Line 27. ‘migrate’ ‘delay’ ‘take’ - should these verbs be past tense since they reflect post-analysis data?

Line 33. I think it would be appropriate to clarify here that there were five aggregated or multi-stock index populations, rather than just “different populations”.

Line 39. “This suggests that this” is a very ambiguous phrase.

Line 40. My research group and others have advocated for using the term “thermal refuges” rather than “thermal refugia”. The latter term is generally used in reference to much larger geographic scales than the small tributary confluences described in the manuscript.

Line 49. Another case where an ambiguous “This” begins a sentence.

Line 51. As noted in general comments, this idea about protecting habitats does not really reemerge in the Discussion section.

Line 69. Consider rephrasing. “Basins” do not have migrations.

Line 73. Consider adding “U.S.” before Endangered Species Act, since the Columbia River basin includes rivers in Canada.

Line 100. Should “frequency” be replaced with a term like “individual probability” or “likelihood”?

Line 114. Perhaps ‘mid-Columbia River (hereafter mid-Columbia)’

Line 126. Insert “many” before “inland”? There are several summer-run steelhead populations East of the Cascade Crest that do not pass through this reach.

Line 129. Although a zip file with environmental data was provided, the manuscript itself does not very clearly show this interannual variability.

Line 132. It was not clear to me how the presented data could be used by dam managers to mitigate mortality?

Line 133. The authors might want to include some hypotheses in this paragraph. Doing so would help structure and organize the rest of the manuscript.

Line 137. The term “known-origin” applies to both hatchery and wild fish.

Line 141. The term “kelts” should probably be changed to “repeat spawners” because kelts really just refers to downstream-migrating adults.

Line 152. Insert “adult” before “migration timing”.

Line 155. It is not clear to me why the John Day and Umatilla populations are singled out here? Several other summer-run steelhead populations originate downstream from McNary Dam but east of the Cascade Crest, including Deschutes, Wind and Klickitat River, Fifteenmile Creek, and Rock Creek populations.

Figure 2 caption. I recommend defining the reach endpoints here, and clarifying that times were for pit-tagged adult steelhead. In addition, I think the authors refer to the study reach as mid-Columbia, not lower Columbia, elsewhere.

Line 191. Some readers may question why there are~32,000 fish here but ~43,000 in line 143. Presumably, the ~11,000-fish difference mostly reflects mortality given the high detection efficiency at McNary Dam?

Lines 207,212 .’is’ = ‘was’? Past tense for completed work.

Line 233. “Season” is vague in this context. Spring – fall?

Line 235. Perhaps this expectation about variable effects could be restated as a hypothesis in the introduction. Alternately, please provide some rationale for why this was expected.

Line 244. Keefer et al. (2008, Ecol App) and Tattam & Ruzycki (2020, TAFS) also highlighted juvenile barging effects on adult steelhead. See full citations below.

Lines 266, 267. ‘are’ = ‘were’? Also consider changing “as adults” to “in adults”.

Line 279. Rephrase. No fish were released from Bonneville Dam, only detected there.

Line 297. The statement about harvest is somewhat unclear. Does post-release mortality mean fish that were released from recreational fisheries? How were the Jording harvest data estimated and/or vetted? Worth describing because there is considerable agency disagreement about steelhead harvest rates in the study reach.

Line 312. Was migration resumption based solely on Fish detection at McNary Dam?

Line 323. “simulate”= “simulated”. See previous verb tense comments.

Line 341. I recommend that the authors use “daily mortality rate” rather than “mortality rate” in most cases. It is important to establish and maintain that this is a daily estimate to avoid potential confusion with reach survival language.

Line 366. Consider adding “by index population” to the bold text portion of the caption.

Line 381. Consider adding “d” or “days” after 236.

Lines 402, 450. “are”= “were”.

Line 403. “This combination” is somewhat ambiguous because no combination has been mentioned.

Line 434. “delay” = “delayed”. I suggest carefully checking the manuscript for appropriate verb tense usage.

Line 488. “Snake River models” = “models for Snake River index populations”?

Line 495. It might be a good idea to capture this proportional changes idea in the Abstract.

Line 510. “This aligns” - somewhat ambiguous this and questionable verb tense.

Lines 517,521. Insert “daily” before “mortality rates”, as suggested previously.

Line 531. “this period” is somewhat vague given long separation from line 525.

Line 535. Are these r values for pairwise correlations?

Line 548. Typo – delete 2nd ‘migrants’.

Line 579. Consider rephrasing “stressful dam interactions”

Line 589. This is one of the statements that makes me question why other PIT detections (i.e., at upstream dams and tributaries) were not considered in the analysis. See general comments.

Line 603. “Stream”= “River” “is”= “was”?

Line 606. Should the term “annual” be inserted before “reach survival”?

Line 608. Should a term like “instantaneous” be inserted before “impacts”?

Line 627. I think it is important that the authors clarify somewhere in the manuscript that the Zone 6 harvest estimates are for main stem Columbia River fisheries only, these estimates do not include harvest inside most thermal refuge areas, to the best of my knowledge.

Line 643. “Chinook” = “Chinook salmon” and the scientific name should be added here.

Line 657. Add “River” after “Salmon” and “Clearwater”.

Line 697. This statement is not accurate. Lower reaches of both rivers, but especially the Salmon River, can be very warm in late summer and early fall. Clearwater River water temperatures are also manipulated by coldwater releases from Dworskak Dam.

Line 711. Add “River” after “Columbia”.

Line 724. It is worth mentioning that the state of Oregon has recently imposed steelhead fishery closures inside thermal refuge sites.

Line 739. “Stock-specific”

lines 744. Typo: ‘taging’. Reference #32 would be very appropriate here.

Line 750. Keefer & Caudill (2016) documented steelhead use of thermal refuges in the reach between Ice Harbor and Lower Granite dams.

Line 1005. Seems like the word steelhead should appear in this and other captions. I also suggest that the word “Dam” should follow Bonneville and McNary in almost every use.

Lines 1014,1022. Some variables need to be defined in the captions.

Figure 1. In the legend, migration-blocking should be hyphenated.

Figure 5. I question why the temperature scale starts at 18°C, since some steelhead migrating in June, September, and October almost certainly encountered cooler temperatures? Note that figure 6 scale starts at 17° C.

Figure 7. It is not clear to me why the harvest rates were only applied to two index populations?

Figure 8. Panel C shows an interesting pattern. Is it possible that the upswing in mortality rates reflect the timing of fishery effort?

Figure 9. There is a lot of information presented here. Perhaps too much to interpret in a single figure?

Supplementary figures. I found these interesting and useful.

Some potentially useful references to consider:

Hess, M. A., and coauthors. 2016. Migrating adult steelhead utilize a thermal refuge during summer periods with high water temperatures. ICES Journal of Marine Science 73(10):2616-2624.

Keefer, M. L., and C. C. Caudill. 2016. Estimating thermal exposure of adult summer steelhead and fall Chinook salmon migrating in a warm impounded river. Ecology of Freshwater Fish 25:599-611.

Keefer, M. L., C. C. Caudill, C. A. Peery, and S. R. Lee. 2008. Transporting juvenile salmonids around dams impairs adult migration. Ecological Applications 18(8):1888-1900.

Tattam, I. A., and J. R. Ruzycki. 2020. Smolt transportation influences straying of wild and hatchery Snake River steelhead into the John Day River. Transactions of the American Fisheries Society 149:284-297.

6. PLOS authors have the option to publish the peer review history of their article (what does this mean?). If published, this will include your full peer review and any attached files.

Reviewer #1: No

Reviewer #2: **Yes: **Matthew Keefer

---

## [Author Response · Author response to Decision Letter 0]

22 Jan 2021

Response to Editor 

Madison Powell, PhD

Academic Editor

PLOS ONE

Journal Requirements:

Figure names and section headers were edited to match the form requested by journal style guidelines. In addition, some symbols were replaced in the text with the equation editor as the guidelines request. 

2. Our internal editors have looked over your manuscript and determined that it is within the scope of our Freshwater Ecosystems Call for Papers. This collection of papers is headed by a team of Guest Editors for PLOS ONE (https://collections.plos.org/s/freshwater-ecosystems). The Collection will encompass a diverse range of research articles on biodiversity conservation, including freshwater fish ecology. Additional information can be found on our announcement page: https://collections.plos.org/s/freshwater-ecosystems.

If you would like your manuscript to be considered for this collection, please let us know in your cover letter and we will ensure that your paper is treated as if you were responding to this call. If you would prefer to remove your manuscript from collection consideration, please specify this in the cover letter.

Please consider the manuscript for the Freshwater Ecosystems Call for Papers. 

3.We note that [Figure(s) 1] in your submission contain map images which may be copyrighted. All PLOS content is published under the Creative Commons Attribution License (CC BY 4.0), which means that the manuscript, images, and Supporting Information files will be freely available online, and any third party is permitted to access, download, copy, distribute, and use these materials in any way, even commercially, with proper attribution. For these reasons, we cannot publish previously copyrighted maps or satellite images created using proprietary data, such as Google software (Google Maps, Street View, and Earth). For more information, see our copyright guidelines: http://journals.plos.org/plosone/s/licenses-and-copyright.

1. You may seek permission from the original copyright holder of Figure(s) [1] to publish the content specifically under the CC BY 4.0 license. 

We believe that it was the background states layer that may be the copyright issue. We replaced that with a US states layer from the natural earth public domain data access site as suggested by the editor. All other layers used we believe were already in the public domain (USGS WHD dataset). Figure 1 and all other figures were produced by the authors specifically for this manuscript. 

4. We note you have included a table to which you do not refer in the text of your manuscript. Please ensure that you refer to Table 2 in your text; if accepted, production will need this reference to link the reader to the Table.

In the text it is actually the reverse issue. There was a reference to Table 2 but no table 2 was provided. This reference was deleted as the associated reference for S1 Fig was sufficient. 

Response to Reviewers

Reviewer 1

5. Review Comments to the Author

Reviewer #1: This manuscript examines the variation in migration behavior of summer-run steelhead (anadromous rainbow trout, Oncorhynchus mykiss) through the Bonneville-McNary reach using PIT tag data from fish tagged as juveniles from different populations in the middle upper Columbia River basin, associated estimates of mortality, and the association of various environmental factors. Steelhead that migrate through this reach, which exhibits higher water temperatures due to retention time of the water in this lower reach, often seek thermal refugia and delay their migration to upper reaches for spawning. The authors find that populations show both similarities, such as a delay in migration that proceeded the highest annual temperatures and deceased daily mortality risk with delay despite increased cumulative within-reach mortality, and differences, including the temperature threshold at which delay is made, rates of delay, and associated mortality rates. As steelhead in the middle and upper Columbia are with few exceptions ESA-listed, and understanding the natural and anthropogenic factors that influence mortality in this therefore of great interest, I find that this paper is timely, adequately executed, and an insightful contribution to the relevant literature.

I would recommend, at the authors’ discretion,that they consider clarifying their manuscript in one respect, which the authors allude to, but is not in my opinion sufficiently identified. The ultimate question for migrating Columbia River steelhead is one of total fitness (survivorship to spawn, fecundity, and survivorship of their offspring to reproduction). While migration through the lower Columbia reaches and the decision to delay and seek thermal refugia may create some differential in survivorship out of that reach, the survivorship of fish that do not delay but make it through the reach is not secured, as the authors mention, and more importantly even if they do survive to spawning tributary, this does not necessarily guarantee greater total fitness, since the additional components of fitness may also be affected by the choice to continue migrating through adverse conditions. Again, as the authors allude, a better measure of the cost/benefit (“bet hedging” advantage) provided by migration delay would be to measure the actual reproductive success of fish that delayed or continued through, which would be a cumulative result of those choices. While this is clearly beyond the scope of this study, I think it’s important to clarify that the patterns identified herein only represent one aspect of that overall measure.

We agree with the reviewer’s concern that the results from this study offer only a partial picture of the impacts of migration decisions on reproduction and that this should be made very clear in the manuscript. Reviewer 2 expressed some similar concerns regarding the discussion around the consequences of different migration behaviors. We added to the discussion text at the end of section 4.2. The additions more fully describe why we did not present information on detections upstream of McNary and more clearly state that the consequences to survival we describe in the study are a partial picture of fitness consequences of the migration experience. The edited text is as follows: 

“We note that there are in-stream PIT tag detection arrays in many tributaries that could provide further insight into the consequences of different behavioral decisions on migration survival. However, these arrays have low and variable detection efficiencies compared to the dams and some initial explorations have suggested that efficiencies may depend on river flows. Furthermore, steelhead behavior is similarly complicated upstream from the study reach. Fully accounting for these complex movements given the lower detection efficiencies requires large sample sizes, which to date only exist for fish PIT-tagged at Rock Island Dam in the Upper Colombia (46). Consequently, effectively characterizing these complexities was beyond the scope of this manuscript.

Accordingly, it is important to keep in mind that the impacts we describe on migration survival through the study reach represent only one aspect of the total fitness consequences of alternative migration strategies. Stressful conditions and migration delays lead to higher energetic expenditures for fish (47), which may leave fish more susceptible to disease, predation, and other sources of mortality. Energetically costly migrations in salmonids are associated with increased pre-spawn mortality for fish that survive their migration to reach spawning grounds (48,49), though less is written on this subject for steelhead specifically (e.g., 50). In addition, stressful migrations can lead to reduced development of gametes, diminished aggressiveness and longevity to compete for mates and prime redd locations, and lower egg incubation survival (51,52), which may combine to reduce the reproductive fitness of successful spawners. Future studies would need to assess impacts on reproductive success to fully account for the consequences of different migration behaviors. This could potentially be achieved using genetically based parentage analysis, which is already used extensively in the Colombia River Basin (e.g., 53,54).”

We also think that the last paragraph in the conclusions makes this point. 

One other minor concern that I would have wished the authors to have addressed was that it was odd to me that the authors identified three distinct patterns in the data, both conceptually and in the fit of tri-modal models to the migration timing data. I have always inferred the “decisions” to 1) delay migration and seek thermal refugia and 2) overwinter outside of spawning tributary to be distinct, since the first (presumably) implies an active choice to seek a thermal refuge (usually a non-natal tributary), while the latter is likely more often simply pausing active migration because of lower kinetic limits. Moreover, the authors hardly discuss overwintering as a strategy distinct from the “slow” form of delay in the results, and indeed include them as a joint probability in additional models, which I think would rather make sense (in my mind, they are not distinct vis-à-vis initial choices to delay or not). Moreover, just because a fish does not overwinter in this lower Columbia stretch does not mean that it did not overwinter farther up but still outside the spawning tributary, a facet of fitness that is not assessable here. Given this, I wondered why it was necessary to consider these distinct modes at all. 

While only representing a small proportion of fish, overwintering fish in the study reach clearly demonstrated a distinct travel time mode from a statistical standpoint. Thus the mixture models fit better when accounting for this third mode than they did without, and our first priority was to capture the full suite of behaviors as well as possible. Nonetheless, this last mode was not our primary focus, so although we did use the best-fit statistical models, this group was not separated out in most of our analyses, as the reviewer noted. We therefore see no contradiction between the reviewer’s point and what we did. 

We did examine travel time patterns by day of year specifically for overwintering fish in section 2.4 (see Figure S1). However, efforts to explore factors related to the probability of overwintering were not very fruitful and we decided that they weren’t worth presenting to the readers. This was likely primarily due to small sample sizes, but, as the reviewer suggests, it could also be that the overwintering behavior does not respond as directly to temperature. 

We were left with the choice to either include them in subsequent modeling as a joint probability with slow fish or to ignore them. We combined the probabilities for slow behaviors and overwintering because our focus was on the response to summer conditions, and as the reviewer recognized, it is not necessarily the case that those groups are determined at that time. The decision to overwinter may occur separately at a later date once a fish has already delayed. However, this choice had a minimal impact on our results given that overwintering fish only represented ~1% of tagged fish. 

To clarify this decision, in the methods we added: “We combined the probabilities for slow behaviors and overwintering because our focus was on the response to summer conditions, and it is not necessarily the case that those groups are determined at that time. The decision to overwinter may occur separately at a later date once a fish has already delayed. The overwintering group had a limited impact on our results because very few fish (~1%) fell into that category.” In addition, we believe that the augmented discussion regarding the limitations of our study help address this concern.

Relatedly, I wondered, given that despite obvious modes in the raw and log-transformed run-timing data there is considerable overlap in the two distributions associated with the ‘fast’ and ‘slow’ strategies, what the effect of arbitrarily assigning fish to either category based on highest probability (line 201). I wondered if there was a model variation that would allow using assignment probabilities directly rather than a priori assignment, or why the authors did not explore/utilize that.

We recognize there is overlap in the behavior distributions. We actually did use the continuous predicted probability of delaying produced by the mixture models for the covariate probability of delay models. This is described in section 2.5 where we state, “where pDelay is the combined probability of being slow or overwintering for each individual that was output from the mixture models represented by equation 1”. We agree with the reviewer that this approach is superior as it recognizes the level of certainty in the models. We only used the absolute designations in section 2.4 to assess average travel times for each behavior by date. This was admittedly somewhat confusing given the location of the line the reviewer cited (originally line 201) in the migratory behavior designation section (2.3). We moved this line to section 2.4. In addition we added a couple of other minor edits to make it clearer that this only related to this specific section of analysis. 

Reviewer 2

Reviewer #2: General comments.

This was an interesting, well executed study and a well-written manuscript. Overall, I found the results convincing and most of the conclusions appeared to be reasonably well defended. I have three broadly general concerns and a number of mostly minor questions and suggestions (see specific comments below). 

My first concern regards why the authors chose to ignore the extensive additional PIT-tag detection data that were collected for the studied steelhead upstream from McNary Dam? Most readers familiar with the Columbia River basin will recognize that these additional data were available and could have provided considerably more information about the survival questions addressed in the study. 

PIT tag detector arrays in tributaries have much lower detection efficiencies than the dams. Additionally, the detection efficiencies are distinct in each tributary and some explorations have suggested that they are dependent on flow. Thus detection efficiencies would be higher for fast fish that travel during the summer than fish that wait until the late fall or winter to travel at cooler temperatures and higher flows. Given these complexities, combined with a substantial amount of multi-directional movements of fish in the hydrosystem upstream of McNary, we believe that this was too much to address in this manuscript given our focus on behaviors in the mid-Columbia and the amount of results we already had to show. We would like to examine this further in a subsequent analyses to extract further value from this large dataset. We would be happy to discuss with the reviewer their thoughts on possible ways to approach this. However, we note that the energetic consequences of migration behaviors that could impact spawning success and subsequent reproductive success would still not be accounted for even if we were able to track fish perfectly to the spawning grounds. 

We add some discussion of why we didn’t take this step in discussion section 4.2: “We note that there are in-stream PIT tag detection arrays in many tributaries that could provide further insight into the consequences of different behavioral decisions on migration survival. However, these arrays have low and variable detection efficiencies compared to the dams and some initial explorations have suggested that efficiencies may depend on river flows. Furthermore, steelhead behavior is similarly complicated upstream from the study reach. Fully accounting for these complex movements given the lower detection efficiencies requires large sample sizes, which to date only exist for fish PIT-tagged at Rock Island Dam in the Upper Colombia (46). Consequently, effectively characterizing these complexities was beyond the scope of this manuscript.”

We also edited the last paragraph in section 4.2 to describe how it would be necessary to track fish through spawning success and subsequent reproduction to truly understand the full consequences of migration behaviors. 

My second concern is that the authors have identified a daily-scale survival benefit from cool and cold water refuges, but they have provided little to no commentary about the need to protect or restore these habitats in light of projected regional climate warming. Ensuring the persistence of such habitats is central to ongoing management efforts in the basin. 

We agree with the reviewer’s sentiment that the need to protect refuge habitats should be highlighted. To this end, in the first paragraph of the conclusion section where we discuss how this behavioral flexibility may help steelhead adapt to climate change, we add: “However, the benefits of refuge habitats depends on them remaining accessible and cool, despite potential future impacts from continued climate change and landscape changes. It is therefore essential to identify, protect, and restore these important habitats in the Columbia River and in other basins to ensure that they continue to provide resiliency to the salmon and steelhead populations that depend on them (83,84).”

Third, I think the manuscript would be more effective and potentially reach a broader audience if some effort was made to broaden the geographical scope of the messaging. Steelhead are widely distributed and many populations are vulnerable to warming river conditions, especially in the southern portion of the species’ range.

 The added discussion above regarding the importance of protecting refuge habitats was made to apply beyond the Columbia Basin, as it will likely be important in many places. In addition, we added this statement broadening the implications of our results regarding the expected increase in the use of refuge habitats of steelhead with increasing river temperatures: “Given the extensive geographical distribution of steelhead and their exposure to climate impacts (3), it is likely that populations from other watershed will demonstrate similar responses during the adult migration.” 

I appreciate the opportunity to review this work and to see how the authors have built upon some of the early studies on this subject by our research lab.

Sincerely,

Matthew Keefer, University of Idaho

Specific comments.

1. Title. The current title captures the behavior element of the study but neither the reach survival nor instantaneous mortality aspects.

Change to “Environmentally triggered shifts in steelhead migration behavior and consequences for survival in the mid-Columbia River”

2. Animal Research. Although the study authors did not handle any fish in this study, approval was presumably required for the dozens of PIT-tagging studies that made this analysis possible. Please also see next comment.

3. Data Availability. The PIT-tag data used in this study were available only due to years of effort by many state, federal, and tribal agencies. The PTAGIS data-use-policy states:“Data users should contact data contributors to gain context and to ensure that their intended use of the data is appropriate.” It is not clear to me in the disclosures that the authors took this step, though perhaps there is a special arrangement between NOAA-Fisheries and the juvenile steelhead tagging groups? Regardless, I think the author should – at a minimum - include a nod to the participating agencies in the Acknowledgments section and give some lip service to the appropriateness of included data.

Early progress on this work was presented at multiple meetings where steelhead tag coordinators from the IDFG and Columbia River Inter-Tribal Fish Commission were present and their specific comments were discussed in person by L. Crozier. We have also sent a draft copy of this manuscript to the main tag coordinators. 

Added acknowledgement in acknowledgements section: “Thanks to representatives from the Idaho Department of Fish and Game and the Columbia River Inter-Tribal Fish Commission for comments on early versions of this analysis. Finally, the authors would like to thank the numerous state, federal, and tribal agencies whose extensive tagging and monitoring efforts contributed to the PTAGIS database.” 

Line 25. ‘tag data = ‘tag detection data’

Changed

Line 27. ‘migrate’ ‘delay’ ‘take’ - should these verbs be past tense since they reflect post-analysis data?

Changed

Line 33. I think it would be appropriate to clarify here that there were five aggregated or multi-stock index populations, rather than just “different populations”.

Changed to “five aggregated population groups”. 

Line 39. “This suggests that this” is a very ambiguous phrase.

Changed to be more specific “Lower mortality rates suggest that migration delay..”

Line 40. My research group and others have advocated for using the term “thermal refuges” rather than “thermal refugia”. The latter term is generally used in reference to much larger geographic scales than the small tributary confluences described in the manuscript.

Changed to “refuge” 

Line 49. Another case where an ambiguous “This” begins a sentence.

Changed to “These impacts have left”

Line 51. As noted in general comments, this idea about protecting habitats does not really reemerge in the Discussion section.

See above response to general comments.

Line 69. Consider rephrasing. “Basins” do not have migrations.

Changed to “which require long migrations to access”

Line 73. Consider adding “U.S.” before Endangered Species Act, since the Columbia River basin includes rivers in Canada.

Added

Line 100. Should “frequency” be replaced with a term like “individual probability” or “likelihood”?

Replaced with “likelihood”

Line 114. Perhaps ‘mid-Columbia River (hereafter mid-Columbia)’

Changed to suggestion and made consistent throughout paper. 

Line 126. Insert “many” before “inland”? There are several summer-run steelhead populations East of the Cascade Crest that do not pass through this reach.

Added “many”

Line 129. Although a zip file with environmental data was provided, the manuscript itself does not very clearly show this interannual variability.

Figure 9 

Line 132. It was not clear to me how the presented data could be used by dam managers to mitigate mortality?

Removed reference to dam managers

Line 133. The authors might want to include some hypotheses in this paragraph. Doing so would help structure and organize the rest of the manuscript.

We agree that a hypothesis statement might help the flow of the paper. We added the below hypothesis focused paragraph and references the hypothesis in the discussion:

“As past research has suggested, we hypothesized that all steelhead populations would demonstrate increased probabilities of delay during stressful migration conditions, primarily as a response to high temperatures. Nevertheless, we expected population-specific variation in the propensity to delay as those with higher quality holding habitat in natal tributaries, or access to other refuge habitats further upstream, might be less likely to delay in the study reach. In addition, we expected that populations that were less likely to delay would demonstrate lower overall reach mortality. However, because overall reach mortality does not account for time-dependent mortality, it might overestimate the benefits of rapid migration. Therefore, we included a time-dependent mortality rate in our analysis of reach-survival. We hypothesized that mortality rates would differ less between rapid migrants and delayed migrants, and possibly demonstrate the inverse of the relationship compared to overall reach survival.” 

Line 137. The term “known-origin” applies to both hatchery and wild fish.

Moved term to apply to both

Line 141. The term “kelts” should probably be changed to “repeat spawners” because kelts really just refers to downstream-migrating adults.

Changed

Line 152. Insert “adult” before “migration timing”.

Added 

Line 155. It is not clear to me why the John Day and Umatilla populations are singled out here? Several other summer-run steelhead populations originate downstream from McNary Dam but east of the Cascade Crest, including Deschutes, Wind and Klickitat River, Fifteenmile Creek, and Rock Creek populations.

Edited to be more general

Figure 2 caption. I recommend defining the reach endpoints here, and clarifying that times were for pit-tagged adult steelhead. In addition, I think the authors refer to the study reach as mid-Columbia, not lower Columbia, elsewhere.

Edited beginning of caption to “Histogram of mid-Columbia travel times through the study reach from Bonneville to McNary dam”

Line 191. Some readers may question why there are~32,000 fish here but ~43,000 in line 143. Presumably, the ~11,000-fish difference mostly reflects mortality given the high detection efficiency at McNary Dam?

Added “The lower number of fish detected at both dams compared to our entire dataset (n = 43,495) primarily reflects mortality rates as detection efficiencies at McNary Dam were near 100% during the study period.”

Lines 207,212 .’is’ = ‘was’? Past tense for completed work.

Changed

Line 233. “Season” is vague in this context. Spring – fall?

Changed to summer

Line 235. Perhaps this expectation about variable effects could be restated as a hypothesis in the introduction. Alternately, please provide some rationale for why this was expected.

As described above, we added a hypothesis statement. In addition, we added the following rationale for the expected distinct effects of different variables: 

“While temperature directly impacts metabolism and physiological processes, flow and spill may impact migration by altering the physical requirements of dam passage and upstream migration. In contrast, variation in the probability of delay by arrival date might reflect population-specific evolved behavioral thresholds to account for distinct migration routes and the variation in quality of accessible holding habitats upstream and within respective natal tributaries.”

Line 244. Keefer et al. (2008, Ecol App) and Tattam & Ruzycki (2020, TAFS) also highlighted juvenile barging effects on adult steelhead. See full citations below.

Added citations

Lines 266, 267. ‘are’ = ‘were’? Also consider changing “as adults” to “in adults”.

Changed 

Line 279. Rephrase. No fish were released from Bonneville Dam, only detected there.

Changed to “fish never re-detected following detection at Bonneville Dam”

Line 297. The statement about harvest is somewhat unclear. Does post-release mortality mean fish that were released from recreational fisheries? How were the Jording harvest data estimated and/or vetted? Worth describing because there is considerable agency disagreement about steelhead harvest rates in the study reach.

For clarification, we added the following details: “Data for the non-tribal fisheries were aggregated from catch report cards and quality controlled by state agencies (including an estimated delayed mortality of 10% from catch-and-release). Non-tribal fisheries data included the catch of upstream stocks from refuge habitats, including Drano Lake (the mouth of the Little White Salmon River), the lower Wind River, the lower Deschutes River, and the John Day River Arm of John Day Reservoir (43). Data for the tribal fisheries was collected from tribal creel sampling and expanded based on the number of fish tickets for commercial sale (primarily of Chinook salmon) to attempt to represent the total number of fishers.”

Line 312. Was migration resumption based solely on Fish detection at McNary Dam?

Yes, added “as determined by detection at McNary Dam.”

Line 323. “simulate”= “simulated”. See previous verb tense comments.

Changed

Line 341. I recommend that the authors use “daily mortality rate” rather than “mortality rate” in most cases. It is important to establish and maintain that this is a daily estimate to avoid potential confusion with reach survival language.

Changed to consistently be “daily mortality rates”

Line 366. Consider adding “by index population” to the bold text portion of the caption.

Added “by population groups” as this was generally referred to in the text.

Line 381. Consider adding “d” or “days” after 236.

Added “days”

Lines 402, 450. “are”= “were”.

Changed

Line 403. “This combination” is somewhat ambiguous because no combination has been mentioned.

Added, the combination of “slow, fast, and overwintering” migration behaviors 

Line 434. “delay” = “delayed”. I suggest carefully checking the manuscript for appropriate verb tense usage.

Changed and checked

Line 488. “Snake River models” = “models for Snake River index populations”?

Edited to say “models for Snake River population groups”

Line 495. It might be a good idea to capture this proportional changes idea in the Abstract.

Added to abstract

Line 510. “This aligns” - somewhat ambiguous this and questionable verb tense.

Changed to “This was also”

Lines 517,521. Insert “daily” before “mortality rates”, as suggested previously.

Added

Line 531. “this period” is somewhat vague given long separation from line 525.

Changed to “within this date range”

Line 535. Are these r values for pairwise correlations?

Specified as Pearson’s r

Line 548. Typo – delete 2nd ‘migrants’.

Deleted

Line 579. Consider rephrasing “stressful dam interactions”

Changed to “dam passage failures or strenuous ladder ascensions”

Line 589. This is one of the statements that makes me question why other PIT detections (i.e., at upstream dams and tributaries) were not considered in the analysis. See general comments.

See above where we address this in the general reviewer comments.

Line 603. “Stream”= “River” “is”= “was”?

Changed

Line 606. Should the term “annual” be inserted before “reach survival”?

Added

Line 608. Should a term like “instantaneous” be inserted before “impacts”?

Added short-term, not exactly instantaneous

Line 627. I think it is important that the authors clarify somewhere in the manuscript that the Zone 6 harvest estimates are for main stem Columbia River fisheries only, these estimates do not include harvest inside most thermal refuge areas, to the best of my knowledge.

They do to some extent. Reference harvest description addition above. 

Line 643. “Chinook” = “Chinook salmon” and the scientific name should be added here.

Added

Line 657. Add “River” after “Salmon” and “Clearwater”.

Changed

Line 697. This statement is not accurate. Lower reaches of both rivers, but especially the Salmon River, can be very warm in late summer and early fall. Clearwater River water temperatures are also manipulated by coldwater releases from Dworskak Dam.

This statement was edited to only refer to Upper Columbia fish

Line 711. Add “River” after “Columbia”.

This is referring to the population group, and not the river specifically. We thus leave the river out. 

Line 724. It is worth mentioning that the state of Oregon has recently imposed steelhead fishery closures inside thermal refuge sites.

Added, “For example, the state of Oregon has recently imposed steelhead fishery closures inside thermal refuge sites while the state of Washington has not.”

Line 739. “Stock-specific”

Changed

lines 744. Typo: ‘taging’. Reference #32 would be very appropriate here.

Edited and added reference

Line 750. Keefer & Caudill (2016) documented steelhead use of thermal refuges in the reach between Ice Harbor and Lower Granite dams.

Added citation

Line 1005. Seems like the word steelhead should appear in this and other captions. I also suggest that the word “Dam” should follow Bonneville and McNary in almost every use.

Edited

Lines 1014,1022. Some variables need to be defined in the captions.

Added variable descriptions

Figure 1. In the legend, migration-blocking should be hyphenated.

Edited

Figure 5. I question why the temperature scale starts at 18°C, since some steelhead migrating in June, September, and October almost certainly encountered cooler temperatures? Note that figure 6 scale starts at 17° C.

Changed figure 5 scale to start at 17C to match figure 6.

Figure 7. It is not clear to me why the harvest rates were only applied to two index populations?

Because they were dropped in model selection in the earlier arriving populations. Harvest rate was considered in all populations. While harvest is likely higher in the later arriving populations, it seems unlikely that it is inconsequential to the other populations given that the harvest estimates, if accurate, account for a large proportion of the mortality in the reach. This highlights the need to better resolve harvest estimates by population, which we bring up in the discussion. 

Figure 8. Panel C shows an interesting pattern. Is it possible that the upswing in mortality rates reflect the timing of fishery effort?

This is likely a contributing factor. However, note that this pattern is basically the inverse of the pattern seen in the probability of delay seen in Figure 5B. So it seems that fish less frequently delaying migration is a big part of this. We add a mention that an increase in the fisheries may contribute to this upswing where Figure 8C is referenced in the text. 

Figure 9. There is a lot of information presented here. Perhaps too much to interpret in a single figure?

We considered making these two figures or removing a panel. However, we think that all the plots relate to each other and that the story is clearer with all of them together. Accordingly, we decided to keep them as is. 

Supplementary figures. I found these interesting and useful.

Great!

Some potentially useful references to consider:

Hess, M. A., and coauthors. 2016. Migrating adult steelhead utilize a thermal refuge during summer periods with high water temperatures. ICES Journal of Marine Science 73(10):2616-2624.

Keefer, M. L., and C. C. Caudill. 2016. Estimating thermal exposure of adult summer steelhead and fall Chinook salmon migrating in a warm impounded river. Ecology of Freshwater Fish 25:599-611.

Keefer, M. L., C. C. Caudill, C. A. Peery, and S. R. Lee. 2008. Transporting juvenile salmonids around dams impairs adult migration. Ecological Applications 18(8):1888-1900.

Tattam, I. A., and J. R. Ruzycki. 2020. Smolt transportation influences straying of wild and hatchery Snake River steelhead into the John Day River. Transactions of the American Fisheries Society 149:284-297.

We included these citations where appropriate in the existing text. ________________________________________

---

## [Editor Report · Decision Letter 1]

15 Apr 2021

Environmentally triggered shifts in steelhead migration behavior and consequences for survival in the mid-Columbia River

PONE-D-20-33797R1

Dear Dr. Siegel,

We’re pleased to inform you that your manuscript has been judged scientifically suitable for publication and will be formally accepted for publication once it meets all outstanding technical requirements.

Kind regards,

Madison Powell, PhD

Academic Editor

PLOS ONE

Additional Editor Comments (optional):

All edits have been addressed.
---

## [Editor Report · Acceptance letter]

29 Apr 2021

PONE-D-20-33797R1 

Environmentally triggered shifts in steelhead migration behavior and consequences for survival in the mid-Columbia River 

Dear Dr. Siegel:

I'm pleased to inform you that your manuscript has been deemed suitable for publication in PLOS ONE. Congratulations! Your manuscript is now with our production department. 

Kind regards, 

on behalf of

Dr. Madison Powell 

Academic Editor

PLOS ONE